# The nucleolar phase of signal recognition particle assembly

Amani Issa[1], Florence Schlotter[1], Justine Flayac[1], Jing Chen[3], Ludivine Wacheul[3], Manon Philippe[2], Lucas Sardini[1], Lalia Mostefa[1], Franck Vandermoere[4], Edouard Bertrand[2], Céline Verheggen[2], Denis LJ Lafontaine[3], Séverine Massenet[1]

The signal recognition particle is essential for targeting transmembrane and secreted proteins to the endoplasmic reticulum. Remarkably, because they work together in the cytoplasm, the SRP and ribosomes are assembled in the same biomolecular condensate: the nucleolus. How important is the nucleolus for SRP assembly is not known. Using quantitative proteomics, we have investigated the interactomes of SRP components. We reveal that SRP proteins are associated with scores of nucleolar proteins important for ribosome biogenesis and nucleolar structure. Having monitored the subcellular distribution of SRP proteins upon controlled nucleolar disruption, we conclude that an intact organelle is required for their proper localization. Lastly, we have detected two SRP proteins in Cajal bodies, which indicates that previously undocumented steps of SRP assembly may occur in these bodies. This work highlights the importance of a structurally and functionally intact nucleolus for efficient SRP production and suggests that the biogenesis of SRP and ribosomes may be coordinated in the nucleolus by common assembly factors.

## Introduction

The eukaryotic rough ER coordinates the biogenesis, folding, post-translational modifications, and sorting of membrane-associated and secreted proteins. Protein secretion is crucial to maintaining cell compartmentalization and homeostasis. In eukaryotes, about one-third of all proteins are synthesized through the membrane of the ER before being transported to their final destinations. The principal and best-characterized pathway of protein targeting to the ER involves the signal recognition particle (SRP) (Hsieh & Shan, 2021; Kellogg et al, 2021; Pool, 2022). Although its composition and size may vary greatly across evolution, the SRP is a universally conserved, abundant RNP particle in all cells (Luirink et al, 1992; Egea et al, 2005).

In mammals, the SRP consists of one RNA molecule, 7SL RNA, and six proteins: SRP9, SRP14, SRP19, SRP54, SRP68, and SRP72 (Fig 1). SRP9 and SRP14 form heterodimers; this is also the case of SRP68 and SRP72. The NAC heterodimer, composed of NACα and NACβ, associates with all ribosomes and specifically recruits the SRP to ribosomes translating proteins containing a specific ER-targeting N-terminal sequence (Gamerdinger et al, 2019; Hsieh et al, 2020; Jomaa et al, 2022). SRP binding causes a temporary halt in protein synthesis, and the SRP–ribosome nascent chain complex is then targeted to the ER membrane through interaction with the SR receptor (Kobayashi et al, 2018; Wild et al, 2019; Wu et al, 2019; Jomaa et al, 2021). The signal sequence is released from SRP and inserted into the translocon channel. The SR receptor and SRP dissociate, and translation resumes.

Disruption of the SRP results in dysregulation of ER-associated mRNA translation and secretory protein sorting. All SRP components are essential to cell survival, and SRP deficiencies are involved in multiple types of diseases, including hematological disorders (Faoro & Ataide, 2021; Kellogg et al, 2022; Linder et al, 2023). Importantly, hematopoiesis defects have been linked to ribosome biogenesis dysfunction diseases called ribosomopathies (Wong et al, 2011; Raiser et al, 2014; Venturi & Montanaro, 2020). This provides a first hint that ribosome production and SRP biogenesis might be functionally interconnected.

Despite the essential role of the SRP in cells, its mode of assembly remains largely enigmatic. Current models of eukaryotic SRP assembly are largely based on classical RNA biochemistry analysis and examination of the localization of SRP components in cells. Pioneering work performed 20 yr ago indicated that SRP biogenesis occurs at least partly in the nucleolus, and this provides a second hint of a possible connection between SRP and ribosome assembly (Jacobson & Pederson, 1998; Ciufo & Brown, 2000; Politz et al, 2000; Sommerville et al, 2005). Quite surprisingly, despite the groundbreaking nature of these observations, the existence of this putative link has not been investigated further.

Previous studies have indicated that SRP assembly is a sequential process in vivo starting in the nucleoplasm, where the RNA component

---

[1]Université de Lorraine, CNRS, IMoPA, Nancy, France   [2]IGH, University Montpellier, CNRS, Montpellier, France   [3]RNA Molecular Biology, Fonds de la Recherche Scientifique (F.R.S./FNRS), Université libre de Bruxelles (ULB), Charleroi-Gosselies, Belgium   [4]IGF, University Montpellier, CNRS, INSERM, Montpellier, France

Correspondence: severine.massenet@univ-lorraine.fr; denis.lafontaine@ulb.be; celine.verheggen@igh.cnrs.fr

7SL is synthesized by RNA polymerase III (Pol III). Assembly continues in the nucleolus, where five of the six SRP proteins (all but SRP54) assemble with the RNA, for some of them maybe co-transcriptionnaly, and it is finalized in the cytoplasm, where the sixth SRP protein joins the particle to produce the mature SRP (Fig 1) (Jacobson & Pederson, 1998; Ciufo & Brown, 2000; Politz et al, 2000, 2002; Sommerville et al, 2005; Massenet, 2019; Kellogg et al, 2021; Gussakovsky et al, 2023). Only a few trans-acting factors have been implicated in SRP assembly thus far. The final cytoplasmic step of assembly has been shown to involve the "Survival of Motor Neurons" complex (Piazzon et al, 2013). Soon after 7SL synthesis in the nucleoplasm, its polyuridylated 3′ end is bound by La. Then, the last three uridines of 7SL RNA are removed and an adenylic acid residue is added by poly(A) polymerase γ (Sinha et al, 1998; Perumal et al, 2001). In yeast, binding of the La homolog Lhp1 to the RNA is required for accurate RNA processing (Leung et al, 2014).

The nucleolus is a biomolecular condensate (formed by liquid–liquid phase separation) where the initial steps of ribosome biogenesis take place (Lafontaine et al, 2021; Yoneda et al, 2021). Ribosome synthesis is a highly complex process requiring the coordination of hundreds of events leading to the production of the mature 40S subunit (SSU, small subunit) and 60S subunit (LSU, large subunit) (Lafontaine, 2015; Bassler & Hurt, 2019; Klinge & Woolford, 2019; Schneider & Bohnsack, 2023). Several hundred trans-acting factors, both proteins and ribonucleic entities, are involved (Tafforeau et al, 2013; Lafontaine, 2015; Klinge & Woolford, 2019; Schneider & Bohnsack, 2023). The process starts in the nucleolus, where the ribosomal RNA precursor 47S pre-rRNA is first synthesized by RNA Pol I and then modified, folded, and processed to yield the mature 5.8S, 18S, and 28S rRNAs. Eighty ribosomal proteins (r-proteins) are produced in the cytoplasm and imported into the nucleus and nucleolus for packaging with the pre-rRNAs. The 5S rRNA is produced by RNA Pol III in the nucleoplasm. 5S associates with two r-proteins (uL5 and uL18), forming a stable trimeric complex, 5S RNP, which then integrates into the pre-60S to form a remarkable architectural feature of the ribosome: the central protuberance.

The nucleolar structure is dynamically regulated and reflects its function in ribosome biogenesis. This explains how the size of the nucleolus, its shape, and even the number of nucleoli per cell nucleus may vary greatly in both normal processes, such as cell differentiation, and pathological ones, such as tumorigenesis, viral infection, neurodegeneration, aging, and responses to stress. The nucleolus contains three main subcompartments nested like Russian dolls: the fibrillar center (FC), the dense fibrillar component (DFC), and the granular component (GC). A fourth subcompartment has recently been described between the DFC and the GC, called the periphery of the DFC (PDFC) (Shan et al, 2023). Perturbations of ribosome biogenesis often lead to disruption of nucleolar architecture and vice versa (Boulon et al, 2010b; Hernandez-Verdun et al, 2010; Lafontaine et al, 2021). Several proteins are important in promoting the establishment/maintenance of the nucleolar phases. These include ribosomal proteins such as uL5 and uL18 (Nicolas et al, 2016), and ribosome assembly factors such as fibrillarin (FBL), nucleophosmin (NPM1), and the helicase DDX21 (Feric et al, 2016; Yao et al, 2019; Lafontaine et al, 2021; Wu et al, 2021).

In this work, considering that ribosomes and SRPs are destined to work together in the cytoplasm, that their assembly is initiated in the same subcellular compartment (the nucleolus), and that mutations in their components lead to similar hematopoiesis deficiency syndromes,

we have investigated the possibility that their biogenesis might be coordinated, possibly by common trans-acting factors, and rely on a functionally intact nucleolus.

# Results

## Production of reporter SRP cell lines for assembly and subcellular localization studies

To study SRP biogenesis, we used the Flp-In T-REx system to construct stable HEK293 and U2OS cell lines inducibly expressing an SRP protein tagged with a GFP epitope. We focused on the nuclear phase of SRP biogenesis and produced cell lines expressing tagged SRP9, SRP14, SRP19, or SRP72. We confirmed induction of GFP-tagged SRP protein expression by the addition of doxycycline (Dox) to the culture medium and found it was possible to match the expression of each GFP construct with that of its endogenous counterpart by selecting appropriate induction conditions (Fig S1A–H). Interestingly, we detected three bands upon probing Western blots for SRP14 (Figs 2 and S1D, H, and I) and two upon probing for SRP72 (Figs 2 and S1B, F, and J). All these bands corresponded to bona fide forms of the SRP14 or SRP72 protein, as shown by their reduction upon siRNA-mediated knockdown (Fig S1I and J).

Next, we tested whether the GFP-tagged SRP proteins expressed in cells could efficiently assemble into SRPs. The expression of each GFP-tagged SRP construct was induced for 3 h in the appropriate HEK293 cell line, and its association with the other SRP proteins was analyzed by immunoprecipitation (IP). Associated proteins were separated by SDS–PAGE and analyzed by Western blotting (WB) with specific antibodies (Fig 2). As all SRP subunit proteins were efficiently recovered when GFP-SRP19 or GFP-SRP72 was used as bait, it appeared that these two tagged proteins had efficiently assembled into mature SRPs (Fig 2A and B). Interestingly, only the short form of SRP14 was pulled down. This indicates that this form is the one that predominantly assembles into mature SRPs. The different forms of SRP14 may be differentially modified versions of the protein or may arise through the translation of alternatively spliced transcripts. In contrast, both forms of SRP72 were efficiently immunoprecipitated, which indicates that they were both assembled into mature SRPs. To determine the contribution of RNAs in these associations, we repeated the co-precipitation analysis in the presence and absence of RNase A (Fig 2A and B). Treating cell lysates with RNase A disrupted or strongly reduced the interactions between proteins binding to the Alu segment of 7SL (SRP9 and SRP14) and those binding to the S domain (SRP19, SRP54, SRP68, and SRP72), as previously reported for mature SRP (Gundelfinger et al, 1983). We conclude that GFP-SRP19 and GFP-SRP72 can each be well packaged into mature SRPs.

In contrast, when we conducted similar analyses using GFP-SRP9 or GFP-SRP14 as bait, we found each to associate almost exclusively with its direct binding partner (tagged SRP9 with SRP14, and tagged SRP14 with SRP9) (Figs 1 and 2C and D). We conclude that the two tagged proteins are present in cells principally in the form of SRP9/SRP14 heterodimers, which implies that they are mostly not incorporated faithfully into mature SRPs. With a more sensitive assay (metabolic labeling/SILAC), however, it was subsequently shown that

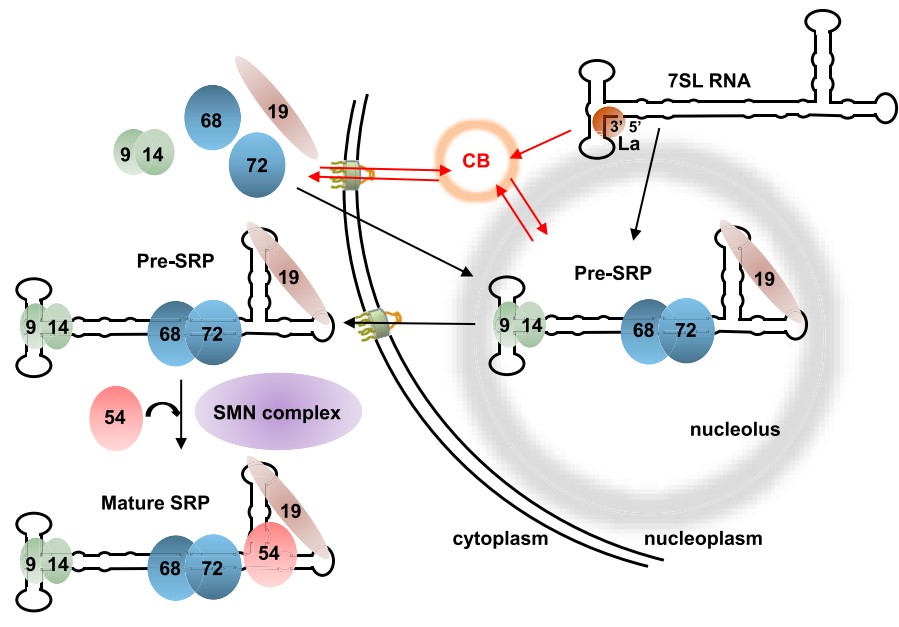

**Figure 1.  Assembly pathway of mammalian SRP.**
The RNA component of SRP, 7SL, is synthesized in the nucleoplasm by RNA polymerase III, where its 3′ end is bound by protein La. Other maturation steps occur in the nucleolus and/or putatively the Cajal bodies (CBs) as indicated. Five of the six SRP protein subunits (SRP9, SRP14, SRP19, SRP68, and SRP72) are assembled in the nucleolus (see text for details). After export to the cytoplasm, the sixth subunit (SRP54) is assembled, aided by the SMN complex, and mature SRP is produced. The Alu and S domains of 7SL are indicated.

there is some residual level of assembly, although it is much reduced (see Fig 5C and D). As discussed below, capturing SRPs "stalled" in the assembly pathway offers an additional opportunity to approach experimentally assembly steps that are otherwise inaccessible to biochemical characterization. Note that the results obtained with all expressed SRP constructs were similar in the two cell lines tested here (HEK293 and U2OS) (Figs 2A–D and S2A–D, respectively).

### GFP-tagged SRP proteins localize to nucleoli and Cajal bodies

Next, we studied the subcellular distribution of the GFP-tagged SRP proteins in the U2OS and HEK293 cell lines (Figs 3 and S3). As an important precaution, we made all our observations under conditions where the GFP-tagged proteins were present at levels comparable to those of the corresponding endogenous proteins, as established by WB (Fig S1).

We found GFP-SRP19 and GFP-SRP72 to display almost identical localization patterns: both proteins localized to the cytoplasm (where mature SRPs function) and the nucleoplasm; they also strongly accumulated in nuclear foci reminiscent of nucleoli (Fig 3A using confocal microscopy, and Fig S3A and B using fluorescence microscopy, for U2OS and HEK293 cells, respectively). To establish the nature of the nuclear foci, we examined the co-localization of GFP-SRP19 and GFP-SRP72 with (1) RPA194 (RNA polymerase I subunit), a marker of the FC; (2) FBL, a marker of the DFC; (3) NCL (nucleolin) and URB1, markers of the PDFC; and (4) NST (nucleostemin, GNL3) and PES1, markers of the GC (Fig 3B and C) using confocal microscopy. Both GFP-SRP19 and GFP-SRP72 were found to co-localize very well with NCL and URB1 and to distribute more broadly into parts of the DFC and GC territories. We conclude that they concentrate in the nucleolus and most markedly in the recently discovered PDFC (Lafontaine, 2023; Shan et al, 2023).

In parallel, we also performed co-localization assays with coilin, which labels Cajal bodies (CBs) (Figs 4A and B and S3G and H for U2OS and HEK293 cells, respectively). This confirmed SRP localization to the periphery of the nucleolar DFC (NCL co-staining in the PDFC) and revealed co-localization of both GFP-SRP19 and GFP-SRP72 with coilin. Specifically, analysis of 95 U2OS cells expressing GFP-SRP19 and 45 U2OS cells expressing GFP-SRP72 revealed the presence of these two proteins in about 50% of all analyzed CBs (Fig 4C). Either they are two classes of CBs, containing or not SRP proteins, or SRP proteins are present in all CBs, but the experimental conditions used did not allow to detect SRP proteins in some of them. We conclude that in addition to concentrating in the nucleolus, both GFP-SRP19 and GFP-SRP72 localize to the CBs. This suggests that an as yet unreported step of SRP assembly might occur in CBs (see Fig 1).

Although GFP-SRP9 and GFP-SRP14 appear to assemble unfaithfully, we were still interested in determining their subcellular localization. In both cell lines, we found GFP-SRP9 mostly in the nucleus, with faint cytoplasmic staining (Fig S3C and D). In agreement with the fact that GFP-SRP9 associates almost exclusively with SRP14 (Fig 2C), the GFP-SRP9/SRP14 dimer appeared to accumulate in the cell nucleoplasm. Conversely, GFP-SRP14, which associates almost exclusively with SRP9 (Fig 2D), accumulated throughout the cytoplasm at a higher level than GFP-SRP9 and gave rise to faint nucleoplasmic staining (Fig S3E and F). Co-localization experiments with NCL showed GFP-SRP14 to accumulate in the nucleoli, albeit to a lesser degree than GFP-SRP19 or GFP-SRP72 (Fig 4D). In this case, no co-localization with coilin was observed. Given the different localizations of GFP-SRP9 and GFP-SRP14, we conclude that adding a GFP tag may interfere differentially with the tagged protein's assembly.

In conclusion, our analysis confirms the association of SRP19 and SRP72 with the nucleolus. It reveals for the first time an accumulation of

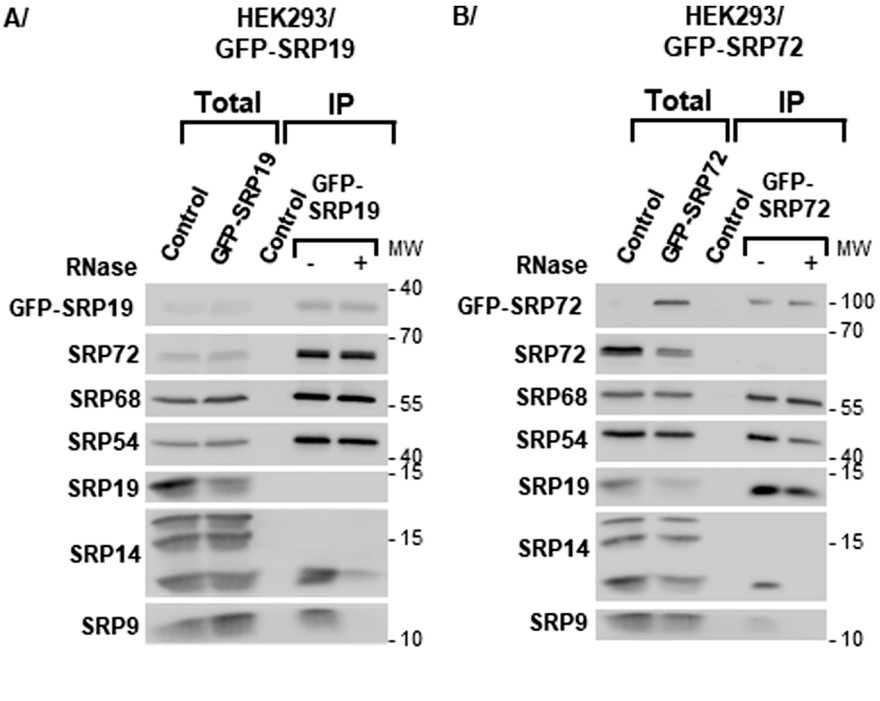

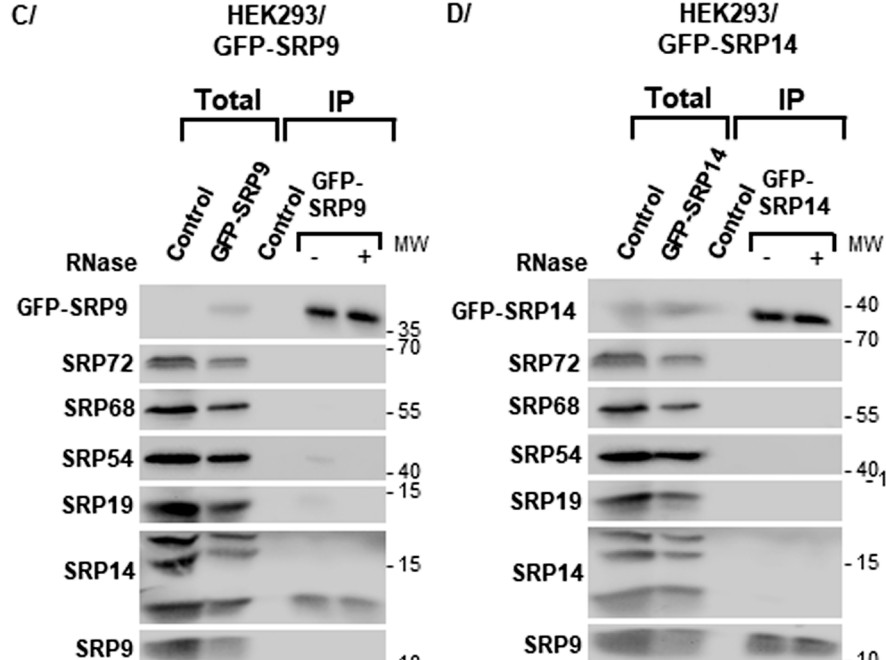

**Figure 2. Analysis of the SRP proteins associated with each expressed GFP-tagged SRP protein.**
**(A, B, C, D)** Total extracts were produced from HEK293 Flp-In T-REx cells having expressed for 3 h one of the following proteins: GFP-SRP19 (A), GFP-SRP72 (B), GFP-SRP9 (C), or GFP-SRP14 (D). IPs were carried out, in the presence (+) and absence (−) of RNase A, with GFP-Trap beads and either one of these extracts or an extract of parental HEK293 Flp-In T-REx cells (Control). The immunoprecipitate (IP) and a fraction of the total cell extract (5%) (Total) were analyzed by SDS–PAGE and WB with antibodies against the indicated proteins. The molecular weight ladder (MW) loaded in parallel with the samples is indicated.

SRP components in CBs, suggesting that an unknown assembly step may occur there.

## SRP proteins associate with scores of nucleolar proteins involved in ribosome biogenesis and nucleolar structure

To reveal novel aspects of SRP biogenesis, we established the interactomes of the two tagged components that faithfully assemble into mature SRPs (GFP-SRP19 and GFP-SRP72) and of the two components stalled in assembly as heterodimers (GFP-SRP9 and GFP-SRP14) (see Fig 2). We found it particularly interesting to characterize the composition of stalled complexes, as this might give us access to specific assembly factors that would otherwise escape identification owing to the transient nature of their intervention. Furthermore, extra-SRP roles have been reported for SRP proteins, just as several ribosomal proteins have been shown to

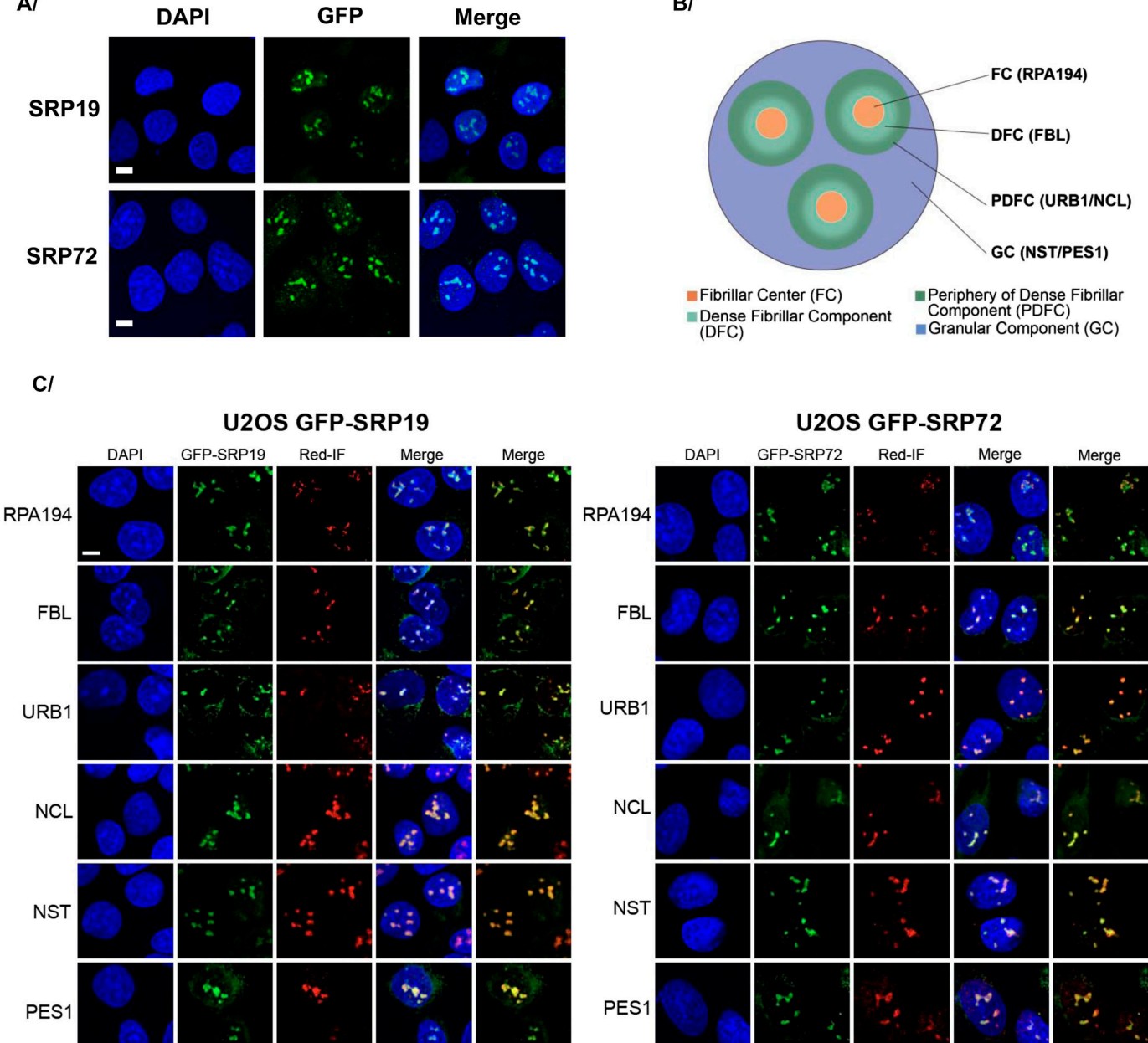

**Figure 3.  Subcellular localization of the GFP-SRP proteins.**
**(A)** Subcellular distribution of GFP-SRP19 and GFP-SRP72 in U2OS cells after 12 h of induction. Direct detection of GFP fluorescence by an Airyscan confocal microscopy. DNA was counterstained with 4',6-diamidino-2-phenylindole (DAPI). Scale bar: 5 μm. **(B)** Schematics describing the main layers of the nucleolus. **(C)** Co-localization studies in U2OS cells expressing GFP-SRP19 (left) or GFP-SRP72 (right). Images acquired by a spinning disk confocal microscopy. Specific antibodies were used to detect RPA194 (labels the FC subcompartment), FBL (DFC), URB1 and NCL (PDFC), NST, and PES1 (GC) (see the Materials and Methods section for details). Scale bar: 5 μm.

exert regulatory functions outside the ribosome (Warner & McIntosh, 2009; Faoro & Ataide, 2021). Candidates for an extra-SRP role include SRP9 and SRP14: they might bind other Alu RNA elements outside the SRP, as these elements are particularly abundant in cells playing important regulatory roles (Bovia et al, 1995, 1997; Chang et al, 1996; Faoro & Ataide, 2021). It is thus interesting to purify those SRP protein subunits even outside the context of intact SRPs.

Using isotope labeling by amino acids in cell culture (SILAC), we performed a proteomic experiment with either GFP-SRP9, GFP-SRP14, GFP-SRP19, or GFP-SRP72 as bait. The GFP tag is an ideal epitope for such work, as it is known to exhibit minimal nonspecific binding to mammalian cell proteins, as compared to other tags, and to be very efficient in quantitative IP-SILAC experiments (Trinkle-Mulcahy et al, 2008). For SILAC analysis, after labeling of cells expressing GFP-tagged SRP and control cells with differently

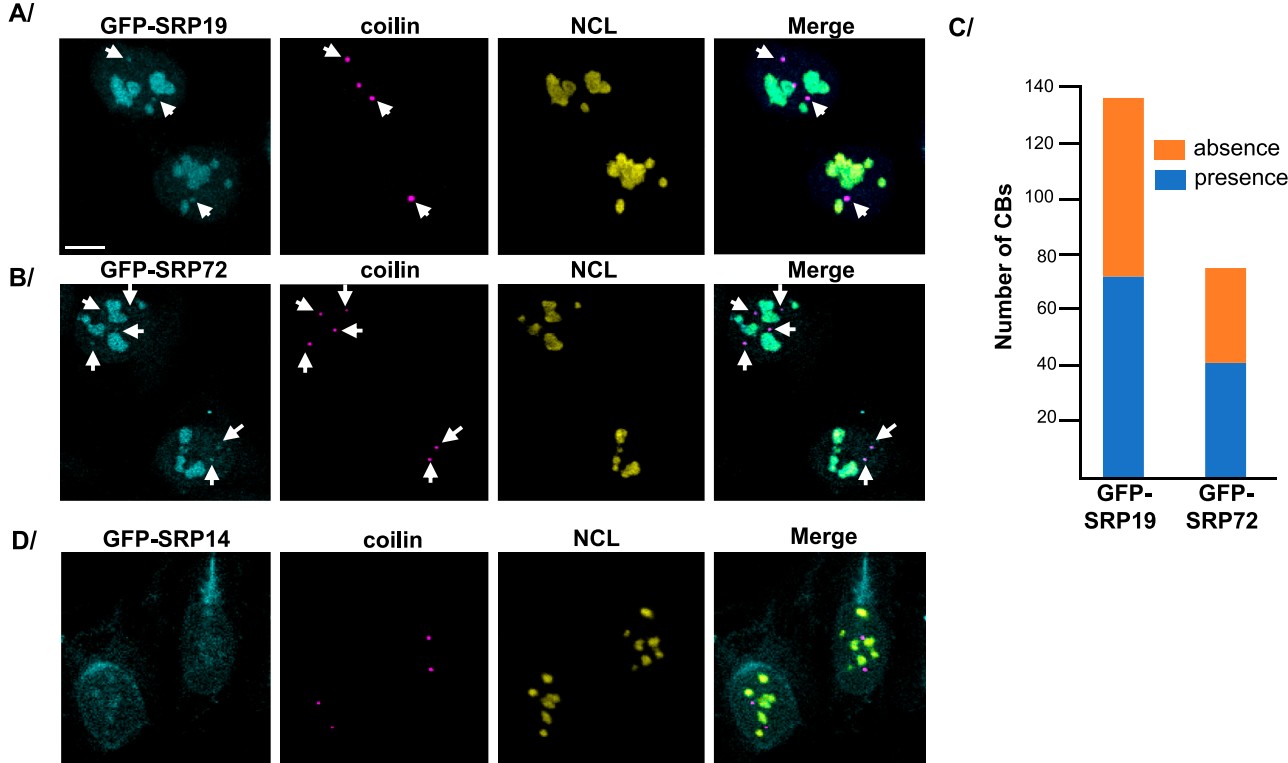

**Figure 4. Detection of SRP proteins in Cajal bodies.**
**(A, B, D)** Expression of GFP-SRP19 (A), GFP-SRP72 (B), or GFP-SRP14 (D) was induced in U2OS Flp-In T-REx cells for 12 h. Double IF experiments were performed, using anti-coilin (a marker of CBs) and anti-NCL (a marker of nucleolus) antibodies. Images were acquired with a scanning confocal microscope. GFP-SRP19, GFP-SRP72, and GFP-SRP14 are in cyan and were located by direct GFP fluorescence detection. Coilin is in magenta, and NCL is in yellow. White arrows indicate co-localization of GFP-SRP19 or GFP-SRP72 with coilin. Scale bar: 8 μm. In (C), a graph is shown, representing the number of CBs containing (in blue) or not containing (in orange) GFP-SRP19 or GFP-SRP72, respectively, in 95 U2OS cells expressing GFP-SRP19 or 45 U2OS cells expressing GFP-SRP72. Counting was done manually by operators blinded to the samples observed.

isotopically labeled amino acids, whole-cell extracts were immu-noprecipitated with anti-GFP antibodies, and the pellets were subjected to quantitative mass spectrometry analysis (Fig 5A–D and Table S1).

A key result of these experiments was that all the GFP-SRP proteins tested were found to associate with factors important for ribosome biogenesis and/or nucleolar structure maintenance (Fig 5A–D, Tables S1 and S2). These include factors involved in early and intermediate steps of ribosomal subunit assembly occurring in the nucleolus and the nucleoplasm, respectively, and in late cy-toplasmic steps (e.g., LTV1 and RIOK2). Surprisingly, the nucleolar partners include 5 of the 12 currently known PDFC components: NCL, NIP7, and the RNA helicases DHX9, DDX5, and DDX21. This tallies with our above conclusion that the SRP proteins of the nuclear fraction largely localize to the PDFC (Fig 3). Other ribosomal assembly factors were also found associated with the SRP, including nucleophosmin (NPM1), the GTPase NOG1, the rRNA modification enzyme NOP2 (NSUN1), and proteins whose function in ribosome biogenesis has been characterized (e.g., MRTO4, NPM3) or not (e.g., RBM28, LYRIC, SND1, C7orf50). In the case of SRP9 and SRP14, re-markably, nucleolar partners and/or proteins involved in ribosome biogenesis were particularly abundant, amounting to about 13% of their interactomes (Fig 5C and D). Some nucleolar partners and/or

proteins involved in ribosome biogenesis were found to be shared by all four SRP baits; others appeared specific to one or several baits (Fig 6A, Table S2A). For the selection of nucleolar preys, we confirmed their association with the endogenous SRP9 and SRP19 proteins by performing pull-downs with specific antibodies and detection of the co-immunoprecipitated proteins by WB (Fig 6B).

Importantly, our identification of interactants was validated technically by the observation that we systematically recovered the expected SRP partner components in our affinity purification. Specifically, we found (1) GFP-SRP19 and GFP-SRP72 to associate with all SRP proteins (Fig 5A and B); (2) GFP-SRP14 to associate tightly and very specifically with SRP9, which confirms its prefer-ential existence as an SRP9/GFP-SRP14 heterodimer (Figs 2 and 5D); and (3) GFP-SRP9 to associate strongly with SRP14 and much less with the other SRP proteins (Fig 5C).

In addition, numerous non-SRP proteins were found to associate with GFP-SRP proteins, and some of them, such as ribosomal proteins, with high affinity and specificity. In particular, GFP-SRP72 and GFP-SRP19 were found to associate strongly with proteins already described as SRP interactors: the two subunits of the SRP receptor (SRα and SRβ), NACα and NACβ, the La protein, and ER proteins (such as LRRC59, CKAP4, MOGS). GFP-SRP9 and GFP-SRP14 were also found to associate with these proteins, but less efficiently

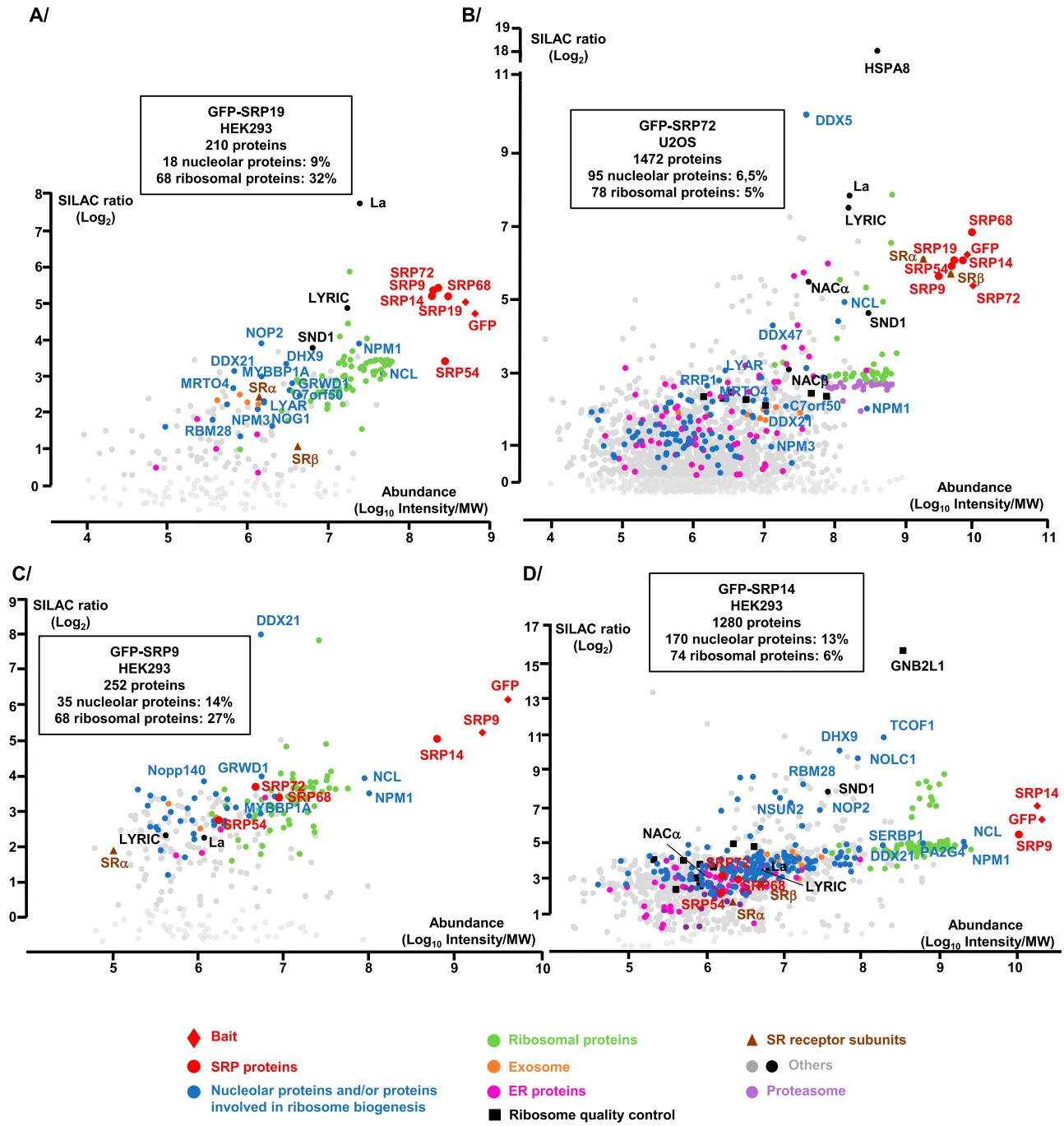

**Figure 5. Proteomic analysis of the partners of GFP-SRP proteins.**
**(A, B, C, D)** IP-SILAC analyses performed on HEK293 Flp-In T-REx cells expressing GFP-SRP19 (A), GFP-SRP9 (C), or GFP-SRP14 (D) for 3 h or on U2OS Flp-In T-REx cells expressing GFP-SRP72 for 3 h (B). The graph displays the log₂ of the SILAC ratio (y-axis, specific IP versus control IP performed with parental Flp-In T-REx cells) as a function of signal abundance (x-axis, log₁₀(intensity)/MW). Each dot represents a protein. The labeled dots were arbitrarily selected to highlight proteins relevant to this study and families of proteins (see Key below the graphs) associated with GFP-SRP proteins. Analysis of the functions of the associated proteins was performed with the Gene Ontology Resource and UniProt. The full hit list with Significance B values is given in Table S1. The indicated percentage of nucleolar proteins and/or proteins involved in ribosome biogenesis, as well as the one of ribosomal proteins, represents the percentages in the number of these classes of proteins among all the associated proteins with the GFP-SRP protein analyzed and with a SILAC ratio above 1.

and with less specificity than GFP-SRP19 and GFP-SRP72. This is in keeping with the fact that GFP-SRP9 and GFP-SRP14 did not assemble into mature SRPs. The experiment further highlighted two new proteins strongly associated with GFP-SRP19 and GFP-SRP72: LYRIC (AEG-1, metadherin) and SND1 (Tudor-SN, p100, EBNA-2 coactivator). SND1 is a major LYRIC-interacting partner, and these

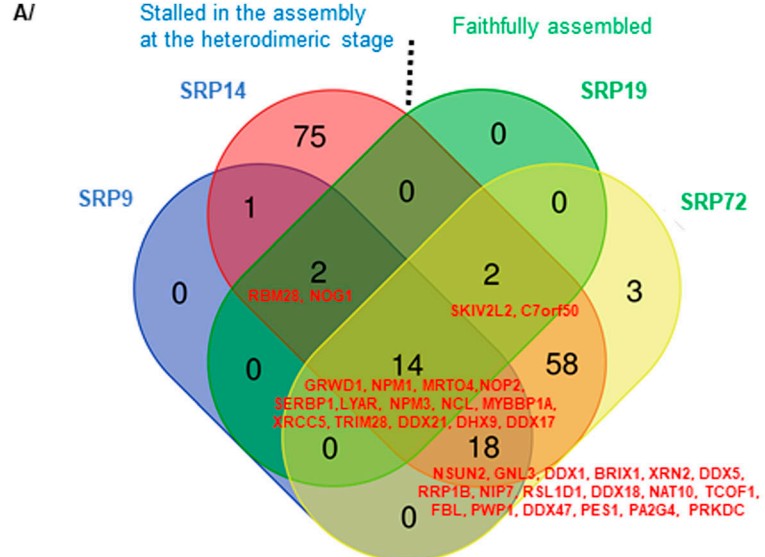

**Figure 6.  Nucleolar proteins associated with GFP-SRP proteins.**
**(A)** Venn diagram showing the intersection between nucleolar proteins and/or proteins involved in ribosome biogenesis present in the GFP-SRP9, GFP-SRP14, GFP-SRP19, and GFP-SRP72 interactomes as determined by IP-SILAC analysis. The diagram includes all the proteins associated with a SILAC ratio up to 1. The ones associated with at least 3 GFP-SRP proteins are listed in red. **(B)** IPs were carried out on U2OS Flp-In T-REx cell total extracts, using anti-SRP19 (upper panels) and anti-SRP9 (lower panels) antibodies bound to magnetic beads with recombinant protein A (Dynabeads Protein A). Beads alone were used as a negative control (Control). The immunoprecipitate (IP) and a fraction of the total cell extract (5%) (Total) were analyzed by SDS–PAGE. The indicated proteins were revealed by WB. The molecular weight ladder (MW) loaded in parallel with the samples is indicated.

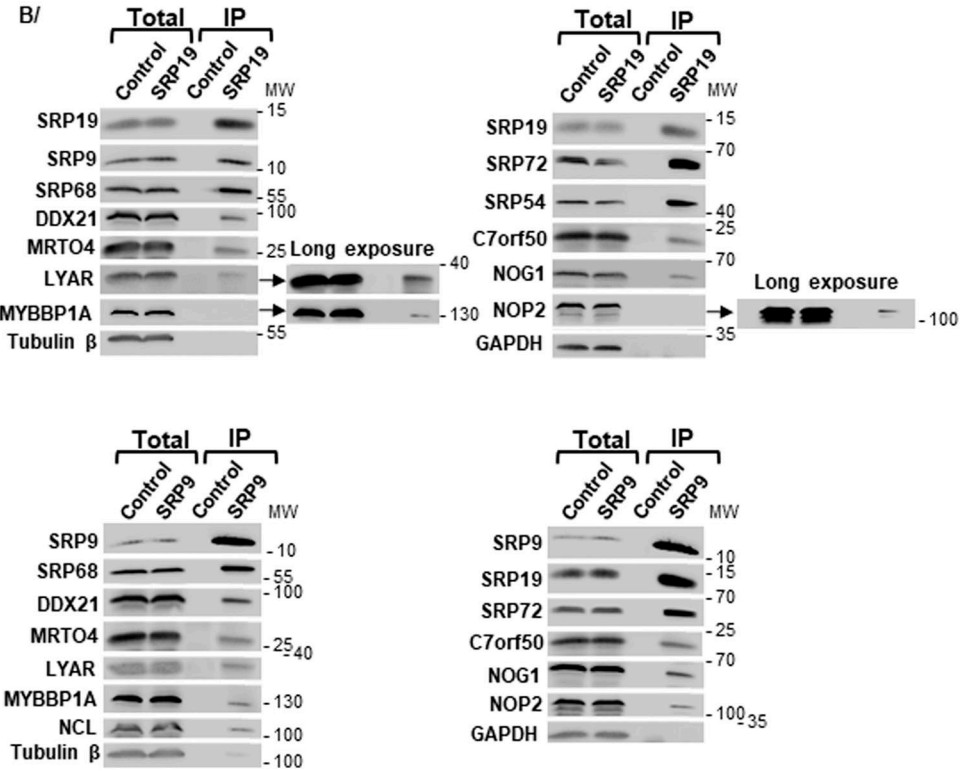

proteins seem to be involved in multiple processes (Abdel Ghafar & Soliman, 2022; Shen et al, 2022a, 2022b). SND1 and LYRIC have previously been located in the cytoplasm and the nucleus, and as nucleolar and ER-associated proteins, depending on the state of the cells studied and on the antibodies used in immunofluorescence experiments (Saarikettu et al, 2010; Gutierrez-Beltran et al, 2016; Wang et al, 2020; Abdel Ghafar & Soliman, 2022). In both cell lines used here, we detected these two proteins mostly in the ER, because they co-localized with the ER protein BiP (Fig S4 for U2OS cells, and data not shown for HEK293 cells). We cannot exclude, however, that they might also be present in other cell compartments.

In conclusion, our proteomic analysis was largely validated by the presence of expected partners, including core SRP components, known assembly factors (La), and ligands (NAC). It expands considerably the notion that SRP interacts with nucleolar proteins and ribosome biogenesis factors by confirming a few known interactions and revealing many novel ones (Fig 9). Among these SRP partners, nucleolar proteins involved in nucleolar structure and/or in early, intermediate, and late steps of ribosome biogenesis. These observations suggest that SRP biogenesis may be intricately linked to the nucleolar structure and ribosome biogenesis. Furthermore, we have uncovered LYRIC and SND1 as novel high-affinity SRP binders. This suggests that they may play a role as SRP assembly factors or in SRP function.

### The nucleolar phase behavior of the SRP is altered upon nucleolar dysfunction

Considering the physical presence of SRP subunits in the nucleolus and the association of SRP subunits with nucleolar proteins (including known ribosome assembly factors), we wondered whether the nucleolar function is required for the association of SRP components with the nucleolus.

To test this, we targeted ribosome biogenesis in GFP-SRP19– and GFP-SRP72–expressing cells using low-dose actinomycin D (Act-D) to inhibit pre-rRNA synthesis by RNA polymerase I (Bensaude, 2011). We then assessed the distribution of GFP-SRP19 and GFP-SRP72 in co-localization experiments with the DFC marker FBL and the GC marker nucleophosmin (NPM1) (Fig 7).

Treating cells with low-dose Act-D is well known to lead to "nucleolar segregation," whereby DFC proteins segregate away from the GC, forming so-called "nucleolar caps" (visible for FBL in Fig 7B and C, "Act-D" condition; see the white arrow in panel B) (discussed in Lafontaine et al [2021]). Under these conditions, most GC proteins redistribute throughout the nucleoplasm (visible for NPM1 in Fig 7D and E, "Act-D"). In the nucleoli of control cells (Ctrl), both of the tested tagged SRP proteins concentrated in irregular zones with rugged contours. This changed strikingly upon Act-D treatment: the distribution of GFP-SRP19 or GFP-SRP72 appeared more compact, into almost spherical zones with abutted FBL caps (Fig 7).

We conclude that upon loss of pre-rRNA synthesis, a core function of the nucleolus, the analyzed SRP proteins relocate differently: whereas FBL forms caps and NPM1 redistributes throughout the nucleoplasm as expected, GFP-SRP19 and GFP-SRP72 distribute into compact spheres, instead of the initial irregular zones with bumpy contours (Fig 7A).

To assess the specificity of this differential redistribution of SRP proteins, we repeated the analysis with six selected nucleolar proteins identified above as SRP partners: two additional PDFC proteins (DDX21 and NCL) and others (MYBBP1A, NOG1, NPM3, and NOP2). Most of them lost their nucleolar association upon Act-D treatment and were completely dispersed throughout the nucleoplasm (Fig S5A). Interestingly, NOG1 was only partially released into the nucleoplasm, maintaining some level of co-localization with GFP-SRP19 (Fig 7F). In this example, the distinct effects of Act-D treatment on NOG1 and SRP19 distribution are particularly obvious.

We have shown previously that only a few of the 80 ribosomal proteins appear so essential to nucleolar structure that their

depletion really impacts it, the most severe effects being noted upon depletion of uL5 (RPL11) or uL18 (RPL5) (Nicolas et al, 2016). To confirm that maintaining nucleolar structure is important for the proper localization of SRP proteins, GFP-SRP19 and GFP-SRP72 U2OS cells were depleted of uL18 for 72 h using silencers. As a negative control, a nontargeting silencer was used (si-Luc). The efficiency of uL18 depletion was established by immunofluorescence with a specific antibody (Fig 8A and B). The localization of SRP19 and SRP72 was established by comparison with that of nucleophosmin (NPM1). As expected, uL18 depletion strongly impacted the distribution of NPM1, GFP-SRP19, and GFP-SRP72 (Fig 8A–C). By comparison, the depletion of the ribosomal protein uS3, which does not disrupt the nucleolar structure (Nicolas et al, 2016), did not alter the localization of GFP-SRP19 (Fig 8D). Thus, disrupting the nucleolar structure by depleting uL18 leads to abnormal localization of GFP-SRP19 and GFP-SRP72.

Several nucleolar proteins identified as SRP partners in this work are known to be required for nucleolar structure integrity (Perlaky et al, 1993; Ugrinova et al, 2007; Lunardi et al, 2010; Kuroda et al, 2011; Wu et al, 2021). To test their importance for the normal distribution of SRP proteins, several of these proteins were depleted in GFP-SRP19–expressing U2OS cells. Specifically, cells were depleted of MYBBP1A, NOG1, DDX21, NPM3, NOP2, or NCL for 48 h (Fig S5B). Each depletion was found to cause disruption of the nucleolar structure and concomitantly affect the localization of GFP-SRP19. In conclusion, the nucleolar distribution of SRP proteins requires an intact organelle.

## Discussion

### SRP associates with scores of nucleolar proteins involved in ribosome biogenesis

SRP biogenesis occurs partly in the nucleolus (Jacobson & Pederson, 1998; Ciufo & Brown, 2000; Politz et al, 2000; Sommerville et al, 2005), where the initial steps of ribosome biogenesis also take place (Fig 1). The nucleolar accumulation of GFP-SRP14, GFP-SRP19, and GFP-SRP72 observed in our inducible cell lines confirms their transient localization to this nuclear subcompartment during SRP assembly. Because SRPs and ribosomes are destined to work together in protein secretion through the co-translational recruitment of ribosomes to the ER, it might be beneficial for cells to coordinate SRP and ribosome production. This coordination might occur in the nucleolus, require a functionally intact nucleolus, and involve common assembly factors. Our data are compatible with this hypothesis. Specifically, our proteomic analysis of two tagged SRP subunits (SRP19 and SRP72) that can faithfully assemble into mature SRPs and of two tagged SRP subunits (SRP9 and SRP14) that remain stalled in the assembly pathway as heterodimers reveal that SRP proteins associate with numerous nucleolar proteins and ribosome biogenesis factor (Figs 5 and 9A, and Table S2A). Careful database mining revealed that ~¼ of all the associations detected in our work had been observed also in previous high-throughput screens, implying that the large majority (239) are entirely new (Fig 9A and Table S2A for details and references). Our data revealed 95 new nucleolar proteins and/or

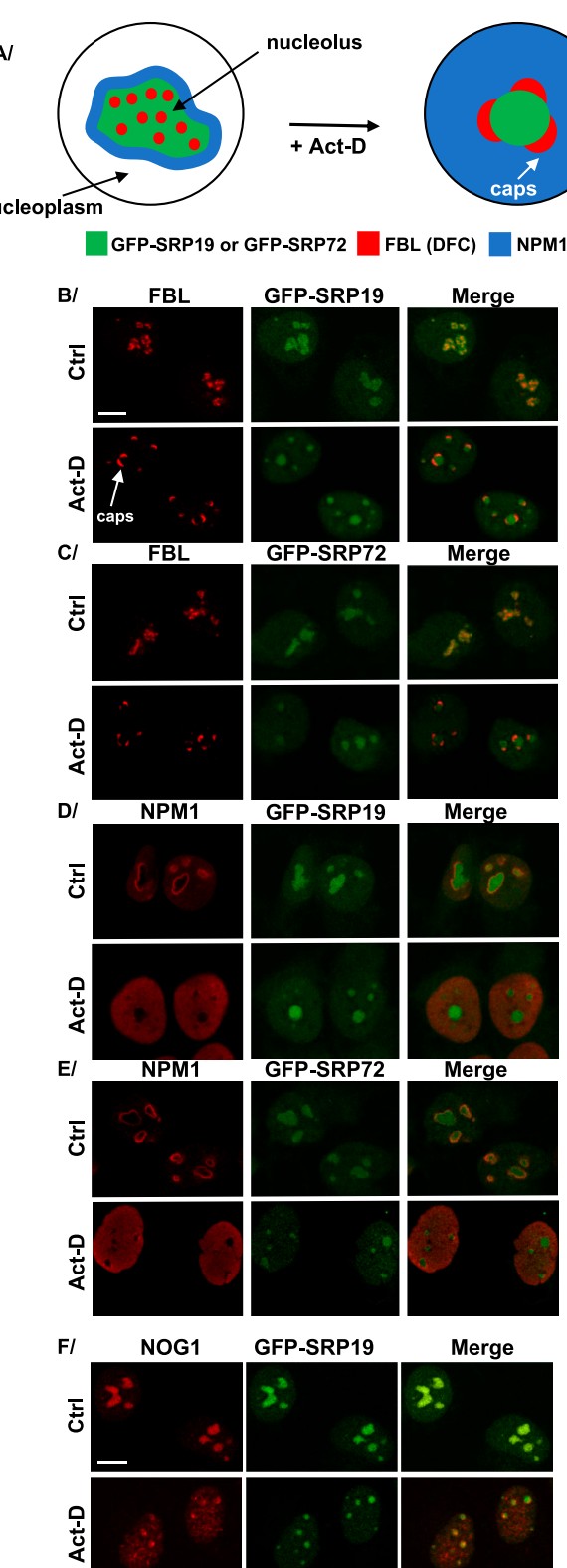

**Figure 7. Functionally intact nucleolus is required for proper localization of SRP proteins.**
**(A)** Schematics illustrating the nucleolar redistribution of GFP-SRP19, GFP-SRP72, FBL, and NPM1 upon actinomycin D (Act-D) treatment. GFP-SRP19 and GFP-SRP72 were relocated from a DFC/GC distribution with rugged contours into a smooth, compact sphere; NPM1 was shifted from a discrete distribution in the GC (lining the periphery) to a distribution throughout the nucleoplasmic space; the distribution of FBL was shifted from bead-like in the DFC to caps. GFP-SRP19 and GFP-SRP72 are in green, FBL in red, and NPM1 in blue. **(B, C, D, E, F)** U2OS Flp-In T-REx cells expressing GFP-SRP19 (B, D, F) or GFP-SRP72 (C, E) for 12 h were treated with Act-D for 2 h. Cells not treated with Act-D were used as negative controls (Ctrl). IF experiments were performed with antibodies against FBL (B, C), NPM1 (D, E), or NOG1 (F). Images were acquired with a scanning confocal microscope. The localization of GFP-SRP19 and GFP-SRP72 (in green) was determined by direct GFP fluorescence analysis. NPM1, FBL, and NOG1 are shown in red. Scale bar: 8 μm (A, B, C, D, E) or 7 μm (F).

proteins involved in ribosome biogenesis that associate with SRP proteins (70 nucleolar proteins involved in ribosome biogenesis, 15 proteins with nucleolar localization but no function in ribosome assembly reporter yet, and 10 proteins involved in ribosome biogenesis in another compartment than the nucleolus) (Table S2A; Fig 9A, proteins circled in red). It brings the number of nucleolar proteins and ribosome biogenesis factors that are SRP interactors to 173. Thus collectively, our work considerably strengthens the notion that SRP is intimately linked to the nucleolus and ribosome production.

Among the SRP partners, we found factors carrying enzymatic activities, for example, GTPases, ATPases, helicases, and rRNA modification enzymes involved in small or large ribosomal subunit assembly, as well as proteins important for nucleolar structure integrity, such as FBL, DDX21, and NPM1 (Tables S1 and S2 and Figs 5, 6, and 9). The interaction of SRP proteins with DDX21 is compatible with a previous report, based on a CLIP assay, indicating that 7SL is the RNA most abundantly associated with this helicase (Calo et al, 2015). DDX21 is part of a recently described subnucleolar domain: the PDFC (Lafontaine, 2023; Shan et al, 2023). Four additional components of the PDFC, that is, nearly half of the known PDFC components, also interact with SRP proteins. This is compatible with the presence of the latter in this region (see Fig 3).

Because we observe that ribosomal assembly factors involved in later steps of subunit biogenesis known to occur in the nucleoplasm, or the cytoplasm, also interact with SRP proteins (Table S2A; proteins in blue circles in Fig 9A), a link between SRP and ribosome biogenesis may extend outside the nucleolus and be even deeper than suggested.

### Tightly regulated SRP production in cells

The idea that SRP production is tightly regulated is suggested by our observation that there is a need to produce the "right amount" of SRP. In our cell lines expressing an exogenous tagged SRP component from a safe harbor locus, we observed that the expression of an additional gene copy encoding a given tagged SRP protein led to a reduced level of the corresponding untagged endogenous protein (see Figs 2 and S1 and S2A–D). This suggests the existence of a regulatory loop preventing the overexpression of SRP proteins. Such a mechanism to control the production of SRP might be lost in cancer cells, where abnormally high levels of 7SL RNA and most SRP proteins have been observed recurrently and in various tumor types, including bladder, breast, colon, liver, lung, prostate, stomach, and thyroid cancers (Faoro & Ataide, 2021; Kellogg et al,

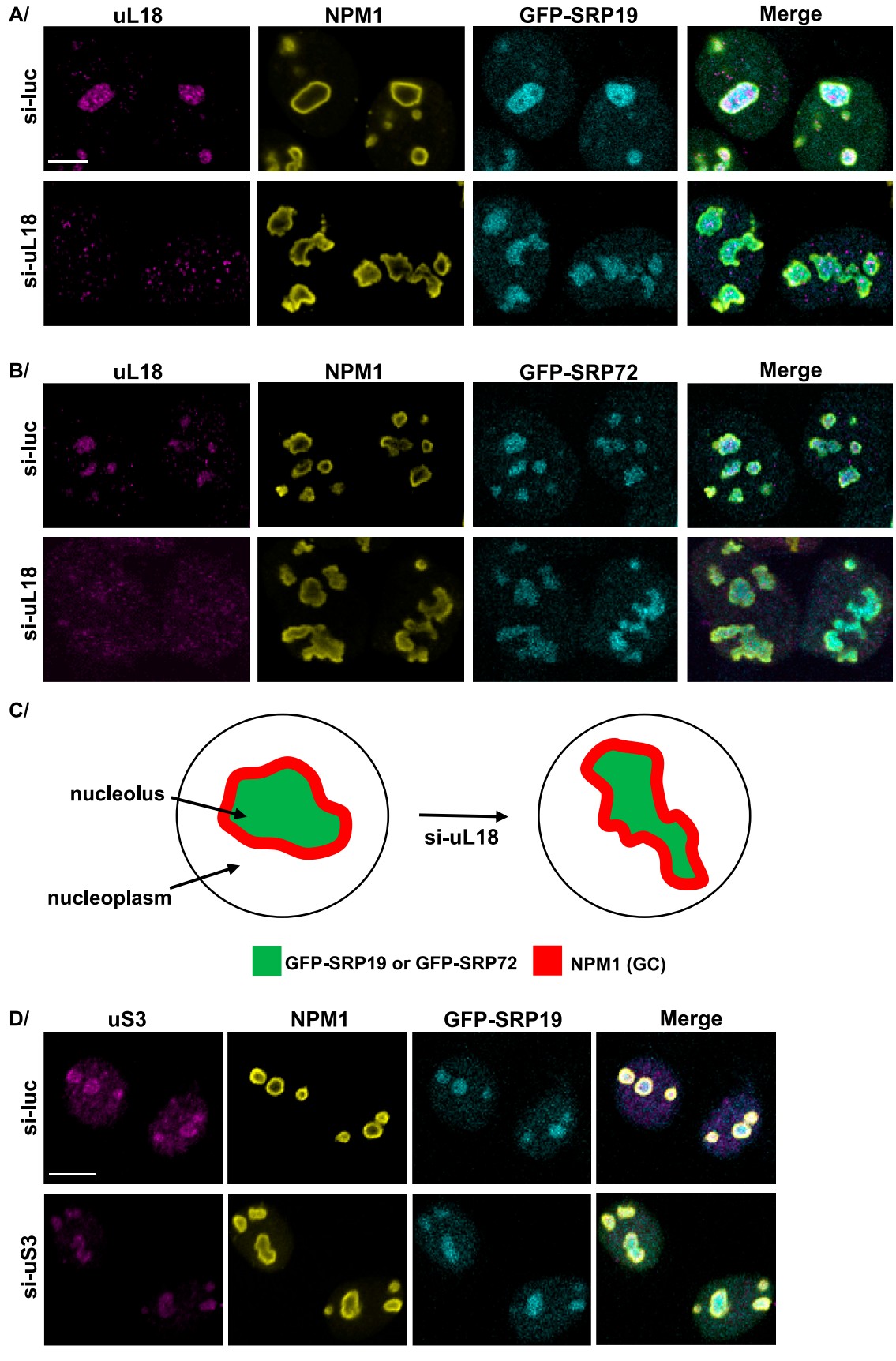

2022). In any case, cancer cells depend heavily on abundant protein synthesis and efficient protein translocation into the ER to sustain their unrestricted growth. Accordingly, components of the Sec complex are also overexpressed in many cancers (Linxweiler et al, 2017; Meng et al, 2021; Müller et al, 2021). Ribosome biogenesis is also well known to be up-regulated in highly proliferating cells (Elhamamsy et al, 2022).

## Our analysis of the SRP interactome reveals novel putative SRP assembly factors

Besides core SRP proteins, ribosomal proteins, ER proteins, nucleolar proteins, and proteins previously known to be associated with the SRP (e.g., La, NACα, and NACβ), our proteomic analysis has revealed other proteins that are associated with SRP proteins (Fig 5, Table S1). Future investigation is needed to test whether any of them are required for SRP biogenesis or function. Among them, we show here that LYRIC and SND1 can bind with very high affinity and specificity to GFP-SRP proteins. Associations of these two proteins with some SRP subunits have already been noticed (Table S2B; Fig 9B), and we are revealing interactions with additional subunits. Neither LYRIC nor SND1 has ever been described as being involved in SRP biogenesis or function. The two proteins work together as a dimer and have been implicated in multiple normal processes and various aspects of tumorigenesis (progression, metastasis, and chemoresistance) (Abdel Ghafar & Soliman, 2022; Shen et al, 2022a, 2022b). Under our experimental conditions, we observed these proteins in the ER (in agreement with Sutherland et al [2004]; Wang et al [2020]), but previously, they have also been detected in the cytoplasm, nucleus, and nucleolus (Saarikettu et al, 2010; Gutierrez-Beltran et al, 2016; Abdel Ghafar & Soliman, 2022). It will be interesting in the future to clarify their role in SRP production and/or function.

## An interface between the SRP and protein quality control?

Interestingly, we found many proteins of the proteasome and the ribosome-associated quality control (RQC) pathway to associate with GFP-SRP72, GFP-SRP14, and GFP-SRP19, such as the E3 ubiquitin–protein ligases listerin and ZNF598, the RNA helicase ASCC3, the ubiquitin-binding protein ASCC2, the ribosome- and tRNA-binding proteins NEMF, TRIP4, and GNB2L1 (Figs 5 and 9C, Tables S1 and S2B). The RQC complex triggers the degradation of aberrant peptides produced by ribosome stagnation and collision, both in the cytosol and at the ER (Phillips & Miller, 2020; Filbeck et al, 2022). We confirmed the association of GNB2L1 with SRP19, and we revealed 20 new associations either with new RQC components or with RQC components that were already known to associate with

other SRP subunits (Fig 9C; Table S2B for details and references). The SRP might play a role in RQC complex recruitment to and/or in its function at the ER. Because the presence of the SRP on mRNA substrates triggers translation elongation arrest, another possibility is that the SRP might inhibit RQC complex function to avoid degradation of nascent peptides destined to be targeted to the ER.

## Loss of nucleolar function leads to altered distribution of SRP19 and SRP72

To disrupt nucleolar function, we treated cells with low-dose Act-D, a specific inhibitor of RNA polymerase I (Bensaude, 2011). This treatment had a marked effect on the subnucleolar distribution of two tested SRP proteins (SRP19 and SRP72), different from its effect on classical markers of other nucleolar subcompartments: the DFC (FBL), PDFC (DDX21 and NCL), and GC (NPM1). Whereas Act-D caused the PDFC and GC markers to leak through the entire nucleoplasmic space, the zones accumulating SRP19 and SRP72 changed shape, losing their initial irregular contours to become rounder and smoother, with directly juxtaposed FBL caps. Severe disruption of nucleolar structure caused by uL18 (RPL5) depletion (Nicolas et al, 2016) also led to the redistribution of GFP-SRP proteins in the disrupted nucleoli. These observations strengthen the view that ribosome biogenesis, nucleolar structure, and the SRP are linked.

## Cajal bodies: a novel site of SRP assembly?

While performing our co-localization studies, we inadvertently found GFP-SRP19 and GFP-SRP72 to accumulate in the CBs in addition to the nucleoli. This unexpected observation is very exciting, as it suggests the existence of additional, CB-located steps of SRP assembly. One limitation of our conclusion is that we used proteins whose genes were expressed from an exogenous promoter. We mitigated this limitation by matching the levels of the tagged proteins with those of their endogenous counterparts. Furthermore, careful re-inspection of "ancient" works performed on the endogenously produced protein made us realize that SRP19 had already been observed in extranucleolar foci some 20 yr ago by the team of Prof. Fried (Dean et al, 2001). Although these authors did not make any claim about this at the time, these extranucleolar SRP19 foci were in all likelihood CBs, as they also contained FBL (FBL is a rare antigen shared by the nucleolus and CBs). From our study and that early work, we conclude that at least some of the SRP proteins transiently reside in CBs. We hypothesize that during SRP biogenesis, these proteins transit through the CBs, where as yet undetermined assembly steps occur (Fig 1). It is known that CBs are the site of several U snRNP assembly events,

**Figure 8. Nuclear localizations of GFP-SRP19 and GFP-SRP72 are disrupted upon uL18 depletion in U2OS Flp-In T-REx cells.**
**(A, B)** U2OS Flp-In T-REx cells expressing the gene encoding GFP-SRP19 (A) or GFP-SRP72 (B) were transfected for 72 h with siRNAs targeting the mRNA encoding the ribosomal protein uL18 (si-uL18). Specific siRNAs against luciferase mRNA (si-Luc) were used as a negative control. The expression of the genes encoding GFP-SRP19 and GFP-SRP72 was induced with Dox for 12 h. Double IF experiments were performed with antibodies against the ribosomal protein uL18 and against NPM1, which marks the GC of the nucleolus. The localizations of GFP-SRP19 and GFP-SRP72 were determined by direct GFP fluorescence analysis. Images were acquired with a confocal microscope. uL18 is in magenta, NPM1 is in yellow, and GFP-SRP19 and GFP-SRP72 are in cyan. Scale bar: 7 μm. **(C)** Schematic representation of the disrupted nucleolus and altered nucleolar distribution of GFP-SRP19 and GFP-SRP72 after uL18 depletion. **(D)** U2OS Flp-In T-REx cells producing the GFP-SRP19 protein were transfected for 72 h with siRNAs targeting the mRNA encoding ribosomal protein uS3 of the small subunit (si-uS3). The legend is the same as in (A, B). Scale bar: 7 μm.

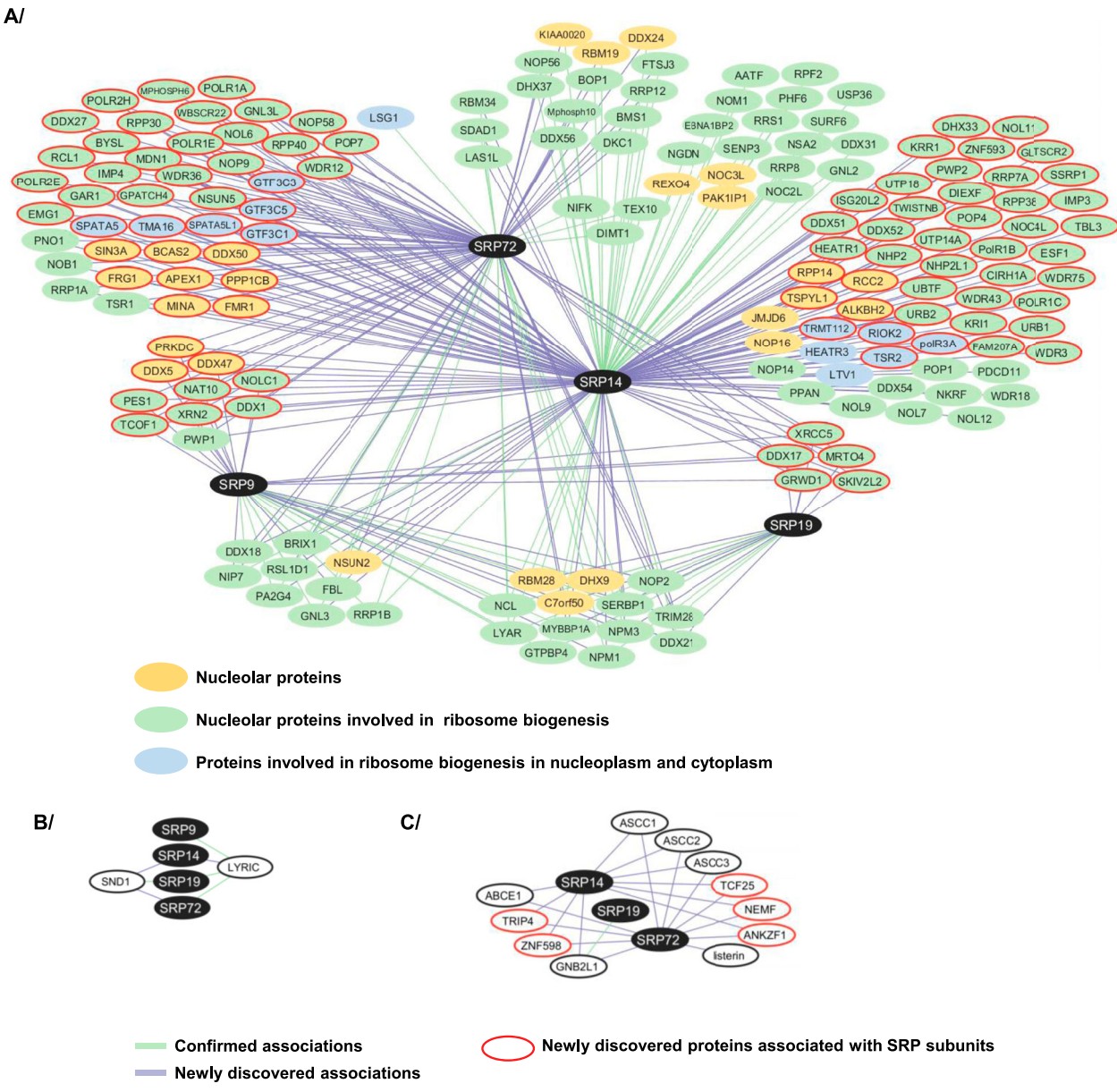

**Figure 9. Network of associations with SRP9, SRP14, SRP19, and SRP72 proteins in humans.**
**(A)** Associations between SRP proteins and nucleolar proteins involved in ribosome biogenesis (in green circles), nucleolar proteins with currently no known function in ribosome biogenesis (in yellow circles), and proteins involved in ribosome biogenesis in the nucleoplasm and cytoplasm (in blue circles). **(B)** Associations between SRP proteins and LYRIC and SND1. **(C)** Associations between SRP proteins and the RQC. The associations uncovered by our data are indicated by a purple line, and the confirmed ones by a green line. The schematics have been prepared with Cytoscape. The previously reported associations were extracted from BioGRID and were described in high-throughput screens (Hayano et al, 2003; Ewing et al, 2007; Todd & Picketts, 2012; Marcon et al, 2014; Hein et al, 2015; Huttlin et al, 2015, 2017, 2021; Kärblane et al, 2015; Boldt et al, 2016; Salvetti et al, 2016; Fasci et al, 2018; Horlbeck et al, 2018; Jang et al, 2018; Liu et al, 2018; Kim et al, 2021; Cho et al, 2022). The new SRP interactors discovered by our data are circled in red.

that is, association of most of the U snRNP-specific proteins with U snRNAs, formation of the U4/U6 di-snRNP and U4/U6-U5 tri-snRNP, and catalysis of U snRNA post-transcriptional modifications by scaRNPs (Will & Luhrmann, 2001; Fischer et al, 2011). The post-transcriptional modification pattern of 7SL RNA is not yet known. Putatively, human 7SL RNA contains several m⁶A and m⁵C residues and one 2'-O-methylated U residue (Khoddami & Cairns, 2013; Gokhale et al, 2016; Dai et al, 2017; Yang et al, 2017; Yue et al,

2018; Chen et al, 2019; Garcia-Campos et al, 2019). Future work will establish whether some of these are formed in CBs and whether one or more steps of SRP assembly occur in these nuclear foci.

In conclusion, by revealing that the interactomes of SRP proteins include numerous ribosome biogenesis factors known to operate in the nucleolus and by demonstrating the importance of a functionally intact nucleolus for proper localization of SRP proteins, our

work has brought to light novel connections between the biogenesis of two essential and functionally related cellular nanomachines: the ribosome and the SRP. We have further highlighted Cajal bodies as a putative novel site of SRP assembly. The evidenced connections between ribosome and SRP biogenesis offer a plausible explanation of why patients harboring SRP mutations display hematological disorder symptoms classically associated with ribosome biogenesis dysfunction.

## Materials and Methods

### Cell cultures, siRNAs, and DNA manipulations

Cells were maintained in DMEM supplemented with 10% FBS, penicillin/streptomycin (10 U/ml), and glutamine (2.9 mg/ml), in a humidified $CO_2$ incubator at 37°C. Stable HEK293 and U2OS cell lines expressing a gene encoding GFP-tagged SRP9, SRP14, SRP19, or SRP72 under the control of a tetracycline-regulated CMV/$TetO_2$ promoter were created with the Flp-In T-REx system (Thermo Fisher Scientific) as recommended by the manufacturer, using the HEK293 or U2OS Flp-In T-REx cell lines and the pcDNA5/FRT/TO plasmid encoding the desired ORF. They were constructed by PCR cloning. Clones were selected in hygromycin B (100 µg/ml), picked individually, and characterized by Western blotting (WB). When required and depending on the experiment and cell type, the expression of the gene encoding a GFP-tagged SRP protein was induced by treatment with 1 µg/ml Dox (D9891; Sigma-Aldrich) for 1–24 h. When indicated, the cells were treated with 0.05 µg/ml of actinomycin D (Act-D) (A9415; Sigma-Aldrich) for 2 h.

When required, the calcium phosphate transfection method was used to transfect cells for either 48 or 72 h before the experiment with 50 nM siRNAs directed against the mRNA coding for the targeted protein (Table S3 for siRNA sequences). Negative control firefly luciferase siRNAs (Gl2) (Elbashir et al, 2002) were used to transfect control cells. The efficiency of siRNA inhibition was tested by WB.

### Antibodies

The following antibodies were used: anti-SRP9 (11195-1-AP; Proteintech) rabbit polyclonal, anti-SRP14 (11528-1-AP; Proteintech) rabbit polyclonal, anti-SRP19 (16033-1-AP; Proteintech) rabbit polyclonal, anti-SRP54 (610940; BD Bioscience) mouse monoclonal, anti-SRP68 (11585-1-AP; Proteintech) rabbit polyclonal, anti-SRP72 (AP17766PU-N; OriGene) rabbit polyclonal, anti-GFP (GTX113617; Genetex) rabbit polyclonal, anti-coilin (ab11822; Abcam) mouse monoclonal, anti-coilin (A303-759A; BETHYL) rabbit polyclonal, anti-NCL (ab136649; Abcam) mouse monoclonal, anti-FBL (72B9) mouse monoclonal (Reimer et al, 1987), anti-NPM1 (ab40696; Abcam) mouse monoclonal, anti-BiP/GRP78 (ab21685; Abcam) rabbit polyclonal, anti-DDX21 (10528-1-AP; Proteintech) rabbit polyclonal, anti-C7orf50 (20797-1-AP; Proteintech) rabbit polyclonal, anti-MYBBP1A (14524-1-AP; Proteintech) rabbit polyclonal, anti-LYRIC (40-6500; Invitrogen) rabbit polyclonal, anti-SND1 (60265-1-Ig;

Proteintech) mouse monoclonal, anti-NPM3 (11960-1-AP; Proteintech) rabbit polyclonal, anti-LYAR (PA5-98969; Invitrogen) rabbit polyclonal, anti-MRTO4 (H00051154-B01P; Thermo Fisher Scientific) mouse polyclonal, anti-NOG1 (GTX110826; Genetex) rabbit polyclonal, anti-NOP2 (10448-1-AP; Proteintech) rabbit polyclonal, anti-uL18 (A303-933A; BETHYL) rabbit polyclonal, anti-uS3 (GTX54720; Genetex) rabbit polyclonal, anti-tubulin $\beta$ (T7816; Sigma-Aldrich) mouse monoclonal, anti-GAPDH (ab8245; Abcam) mouse monoclonal; anti-PES1 and anti-NST (courtesy from E Kremmer); anti-URB1 (PA5-53787; Thermo Fisher Scientific), anti-FBL (ab5821; Abcam), anti-RPA194 (SC-48385; Santa Cruz).

The following secondary antibodies were used: Alexa Fluor 488, 555, 594, or 633 anti-mouse, anti-rabbit, or anti-rat (Invitrogen), secondary antibody coupled to peroxidase (115-035-003; Jackson ImmunoResearch) mouse polyclonal, secondary antibody coupled to peroxidase (A16104; Thermo Fisher Scientific) rabbit polyclonal.

### Immunoblot analysis

Total cell extracts were prepared by resuspending the cell pellet in 0.25 M Tris–HCl, pH 8, buffer and using the freeze–thaw cell lysis method. Cell extracts mixed (vol/vol) with 2x Laemmli buffer were analyzed by 12.5% SDS–PAGE and WB with appropriate antibodies (see above), with the ECL revelation kit (Amersham Biosciences). Systematically, the membranes were cut into pieces according to the molecular weight ladder loaded in parallel with the samples to allow probing of WB for multiple proteins for each experiment. When required, the membranes were stripped according to the manufacturer's protocol (Millipore) and probed a second time with other antibodies. Images were acquired, and quantification was performed with Fusion Solo (Vilber).

### Co-immunoprecipitation and WB

Synthesis of the tagged proteins was induced for 12 h in U2OS Flp-In T-REx cells or 3 h in HEK293 Flp-In T-REx cells, with 1 µg/ml Dox. Total cell extracts were prepared as previously described (Piazzon et al, 2008) in RSB-150 buffer (10 mM Tris–HCl, pH 7.5, 150 mM NaCl, 2.5 mM $MgCl_2$) containing 0.05% NP-40 and incubated for 2 h at 4°C with either (1) specific antibodies bound to magnetic beads with recombinant protein A (Dynabeads Protein A) or (2) GFP-Trap (Chromotech). The beads were washed with the same buffer, and the immunoprecipitated proteins were eluted in Laemmli buffer and analyzed by SDS–PAGE and WB as indicated for immunoblot analysis.

### Immunofluorescence staining (IF) and image acquisition

HEK293 or U2OS Flp-In T-REx cells expressing GFP-SRP proteins were plated respectively at 120,000 or 100,000 cells/well in six-well plates on glass coverslips for 48 h. When required, GFP-SRP protein expression was induced for 12 h before the IF experiments. When indicated, cells were transfected with 50 nM siRNAs or with 1 µg/ml pcDNA5-3xFlag-GFP-LYRIC plasmid, using calcium phosphate for 48 or 72 h before the IF experiment. When indicated, cells were treated with 0.05 µg/ml Act-D for 2 h. Induced cells were fixed in 2% PFA/

PBS 1X for 10 min at RT. After three washes with PBS 1X, cells were permeabilized in 0.5% Triton X-100 for 10 min at RT, then rinsed three times with PBS 1X. Blocking was performed for 30 min at 4°C in PBS 1X containing 3% BSA. Coverslips were then incubated for 2 h at 4°C with primary antibodies diluted in PBS 1X. After rinsing in PBS 1X, coverslips were incubated for 30 min at RT with secondary antibodies coupled to Alexa fluorophore dye (A488, A555, A594, A633; see Antibodies) diluted in PBS 1X. After three washes with PBS 1X and a quick dry, coverslips were mounted on a slide with mounting medium supplemented with DAPI to counterstain the nuclei (Duolink In Situ Mounting Medium with DAPI-DUO82040; Sigma-Aldrich).

For Fig 3A, high-resolution images were captured in Airyscan confocal mode with a Zeiss LSM710 confocal microscope equipped with a 63×/1.4 oil-immersion Plan Apochromat objective.

For Fig 3C, imaging was performed on a Zeiss Axio Observer.Z1 microscope with a motorized stage, driven by MetaMorph (MDS Analytical Technologies) used in confocal mode with a Yokogawa spindisk head, an Evolve camera, a laser bench from Roper (405 nm 100 mW Vortran, 491 nm 50 mW Cobolt Calypso, and 561 nm 50 mW Cobolt Jive), and a 63×/1.4 oil-immersion objective (Plan NeoFluar; Zeiss).

For all the other figures, images were acquired either with a Leica SP5X scanning confocal microscope or with a Nikon epifluorescence microscope. Image analysis and processing were performed with ImageJ.

### SILAC IP and proteomic analysis

SILAC experiments were performed as previously described (Boulon et al, 2010a). HEK293 or U2OS Flp-In T-REx cells inducibly expressing the GFP-tagged proteins were put in 15-cm-diameter plates (six plates per condition) and grown for 15 d in each isotopically labeled medium (CIL/Eurisotop), to ensure complete incorporation of the label. The media were as follows: L-Lysine-$^2$HCl/L-Arginine-HCl light label (K0R0 or condition L, corresponding to the control), L-Lysine-$^2$HCl ($^2$H4, 96–98%)/L-Arginine-HCl ($^{13}$C6, 99%) semi-heavy label (K4R6 or condition M). Control HEK293 or U2OS Flp-In T-REx cells not expressing any GFP fusion protein were cultured under condition L, whereas Flp-In T-REx cells expressing GFP-SRP9, GFP-SRP14, GFP-SRP19, or GFP-SRP72 were cultured under condition M. After 3 h of induction of fusion gene expression by adding 1 $\mu$g/ml Dox to the culture medium, cells were rinsed with PBS, trypsinized, and cryoground. The powder was resuspended in HNT lysis buffer (20 mM Hepes, pH 7.4, 150 mM NaCl, 0.5% Triton X-100, protease inhibitor cocktail [cOmplete; Roche]). Extracts were incubated for 20 min at 4°C and clarified by centrifugation for 10 min at 20,000$g$. For all IP experiments, extracts were precleared by incubation with protein G Sepharose beads (GE Healthcare) for 1 h at 4°C. Each extract was then incubated with 50 $\mu$l of GFP-Trap beads (ChromoTek) for 1.5 h at 4°C, and washed five times with HNT buffer, and beads from the different isotopic conditions were finally pooled. Bound proteins were eluted by adding 1% SDS to the beads and boiling them for 10 min. Proteomic analysis was performed as previously described (Maurizy et al, 2018). MS data were analyzed on MaxQuant software v 2.1.0.0 (Cox & Mann, 2008) with standard parameters and the UniProt database of human canonical protein

isoforms (www.uniprot.org). Proteins were identified with a minimum of two peptides including at least one unique peptide. Relative protein MS intensity (SILAC ratio) was calculated as a median ratio of unique and razor peptide MS intensities. The $P$-value of protein MS intensity enrichment (bait versus control) was calculated as a SILAC ratio M/L, and Significance B according to a method described previously (Cox & Mann, 2008). We have considered as a positive association the associations with a SILAC ratio above 1. The network visualization tool Cytoscape was used to prepare Fig 9 (Shannon et al, 2003).

## Data Availability

The mass spectrometry proteomics data have been deposited to the ProteomeXchange Consortium via the PRIDE (Perez-Riverol et al, 2019) partner repository with the dataset identifiers PXD042191, PXD042192, PXD051488, and PXD042195.

## Supplementary Information

## Acknowledgements

The anti-fibrillarin (72B9) antibody was a kind gift from J. Cavaillé (Toulouse, France), and the U2OS Flp-In T-REx cell line was a kind gift from F-M Boisvert (Sherbrooke, Canada). We thank the MGC collection in Montpellier for providing and preparing plasmids and the PTIBC IBISA (Nancy) and the CMMI (Gosselies) platforms for access to microscopes. Mass spectrometry experiments were carried out within the facilities of the Montpellier Proteomics Platform (PPM, BioCampus Montpellier). We are thankful to S Boulon and B Charpentier for helpful discussions and to C Aigueperse for her help with IF experiments. This work was supported by the French *Centre National de la Recherche* (CNRS), the *Université de Lorraine*, the *Université de Montpellier*, the Lorraine University of Excellence program (LUE), and the *Ligue Nationale Contre le Cancer* (équipe labellisée Montpellier) (grant-30025555 Grand-Est). A Issa is a predoctoral fellow of the Lorraine University of Excellence program (LUE) and L Sardini of the French *Ministère de l'Enseignement Supérieur et de la Recherche*. The authors would also like to thank the ORION program for its contribution to the funding of Lalia Mostefa's research internship (LM). This work has benefited from a government grant managed by the *Agence Nationale de la Recherche* under reference ANR-20-SFRI-0009. Research in the Lafontaine Lab was supported by the Belgian *Fonds de la Recherche Scientifique* (FRS/FNRS), *Université libre de Bruxelles* (ULB), EOS (CD-INFLADIS, 40007512), *Région Wallonne* (SPW EER) Win4SpinOff (RIBOGENESIS), the COST actions EPITRAN (CA16120) and TRANSLACORE (CA21154), and the European Joint Programme on Rare Diseases (EJP-RD) RiboEurope and DBAGeneCure.

### Author Contributions

A Issa: data curation, formal analysis, validation, investigation, visualization, and methodology.
F Schlotter: data curation, formal analysis, validation, investigation, visualization, and methodology.

J Flayac: data curation, formal analysis, validation, investigation, visualization, and methodology.

J Chen: data curation, formal analysis, and investigation.

L Wacheul: data curation and formal analysis.

M Philippe: data curation, formal analysis, validation, investigation, visualization, and methodology.

L Sardini: formal analysis, validation, investigation, visualization, and methodology.

L Mostefa: formal analysis.

F Vandermoere: formal analysis and methodology.

E Bertrand: conceptualization.

C Verheggen: conceptualization, data curation, formal analysis, supervision, funding acquisition, validation, investigation, visualization, methodology, and writing—original draft, review, and editing.

DLJ Lafontaine: conceptualization, data curation, formal analysis, supervision, funding acquisition, and writing—original draft, review, and editing.

S Massenet: conceptualization, data curation, formal analysis, supervision, funding acquisition, validation, investigation, visualization, methodology, project administration, and writing—original draft, review, and editing.

## Conflict of Interest Statement

The authors declare that they have no conflict of interest.

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
