## [Reviewer comments · Life Science Alliance]

Life Science Alliance

The Nucleolar Phase of Signal Recognition Particle Assembly

Amani Issa, Florence Schlotter, Justine Flayac, Jing Chen, Ludivine Wacheul, Manon Philippe, Lucas Sardini, Lalia Mostefa, Franck Vandermoere, Edouard Bertrand, Céline Verheggen, Denis Lafontaine, and Severine Massenet

DOI: <https://doi.org/10.26508/lsa.202402614>

Corresponding author(s): Severine Massenet, University of Lorraine and Denis Lafontaine, FRS/FNRS - ULB

Review Timeline:	Submission Date:	2024-01-23
	Editorial Decision:	2024-02-22
	Revision Received:	2024-05-21
	Editorial Decision:	2024-05-23
	Revision Received:	2024-05-28
	Accepted:	2024-05-28

Transaction Report:

February 22, 2024

Re: Life Science Alliance manuscript #LSA-2024-02614-T

Dr. Severine Massenet
University of Lorraine
CNRS - UMR7365
Avenue de la forêt de Haye
Vandoeuvre-les-Nancy, Cedex 54505
France

Dear Dr. Massenet,

Thank you for submitting your manuscript entitled "The Nucleolar Phase of Signal Recognition Particle Assembly" to Life Science Alliance. The manuscript was assessed by expert reviewers, whose comments are appended to this letter. We invite you to submit a revised manuscript addressing the Reviewer comments.

Thank you for this interesting contribution to Life Science Alliance. We are looking forward to receiving your revised manuscript.

Sincerely,

B. MANUSCRIPT ORGANIZATION AND FORMATTING:

Reviewer #1 (Comments to the Authors (Required)):

Signal Recognition Particle (SRP) targets secreted and transmembrane proteins to the endoplasmic reticulum (ER). SRP assembly is only somewhat understood. The 7SL RNA is transcribed in the nucleoplasm and assembled with a series of protein factors in the nucleolus and cytoplasm as it is assembled into mature SRP in the cytoplasm. SRP partially assembles in the nucleolus and SRP deficiencies are associated with conditions that resemble ribosomopathies, however the relationship between the SRP, the nucleolus, and ribosome biogenesis has not yet been investigated. The authors explored this connection by generating several cell lines that inducibly express GFP-tagged SRP proteins, checking their ability to assemble into SRP particles, determining their subcellular localization, and performing SILAC-based proteomics that revealed association with nucleolar proteins. Further, the authors examined the dynamics of the SRP proteins upon nucleolar disruption.

The authors show that two of the GFP-tagged proteins (SRP19 and SRP72) were able to efficiently assemble into mature SRPs, but two others (SRP9 and SRP14) were not. The latter two were used as representative proteins for complexes stalled in early stages of SRP biogenesis. They found that SRP proteins associate with nucleolar proteins involved in ribosome biogenesis and discovered other novel proteins involved in SRP biogenesis (LYRIC and SND1). They demonstrated that distribution of SRP proteins is altered upon nucleolar disruption with Actinomycin D, and re-discovered definitive co-localization of SRP proteins with Cajal bodies (see Discussion).

Overall, this paper explores the understudied and important connection between the nucleolus and SRP. The method to track the process of assembly with the subset of GFP-tagged SRP proteins stalled at early stages is a unique and interesting approach.

However, I am disappointed in the lack of scholarly use of the published literature on SRP in the nucleolus. For example, it is stated many times that it was "unexpected" that the SRP proteins associate with nucleolar proteins based on the SILAC experiments. They are also referred to as "newly uncovered." Yet the clear association of SRP proteins with nucleolar proteins is already available on numerous databases and has literally been posted there for years (see Biogrid and NCBI gene, for examples.) This information should have been included in the Introduction for a comprehensive review of the literature. Furthermore, a comparison between the results reported here and prior results should be made in the Discussion. How well did prior results report nucleolar association of the SRP proteins? Are the results reported here really an advance, or do they tell us what we already knew but with higher precision because of SILAC? A comparison table should be made and put in the Supplementary Material, and then thoroughly analyzed in the Discussion. This group is not the first to find that nucleolar proteins associate with assembly SRP, and they should not give the impression that they are. It is not right.

Furthermore, the Introduction is too long, and contains extraneous paragraphs on topics that are not touched on further. It should be trimmed by at least a couple of paragraphs (4-5 total).

I understand that it is difficult to probe what we all really want to know: what is the function of SRP assembly in the nucleolus? Given that that is too hard to do, overall the paper represents an important contribution through its exploration of the relationships between SRP, the nucleolus, and ribosome biogenesis if improvements are made as above and below.

Comments:

1. While this paper effectively demonstrate that an intact nucleolus is required for SRP biogenesis, the data do not show that the processes of ribosome biogenesis and SRP are coordinated as stated in the abstract. Can you please soften the language?
2. For proteomics studies, the authors state that a percentage of the total nucleolar proteins (?) associated with the SRPs are nucleolar (see fig. 5). When these numbers were generated, was a cutoff value used to determine a positive association (i.e. the significance value)? The methodology for what constituted a hit and how these percentages were generated should be clarified.
3. Methodology for quantifying imaging data in figure 4 should be explained. How were the number of cells counted chosen? Were they counted manually, and if so by what criteria?

Reviewer #2 (Comments to the Authors (Required)):

The authors report analyses that address the long-standing observation of SRP components within the nucleolus. Tagged constructs are used to follow the localization and assembly status of ectopically expressed SRP proteins. This confirmed nucleolar localization and interactions, and provided evidence for links between nucleolar structure and SRP assembly. In addition, SRP proteins were identified in a subset of CBs, indicating their possible involvement in specific assembly steps.

Overall, the novel insights are limited, but the analyses appear to have been well-performed, while the MS is well written and will be of value and interest to the field. I would support publication with only minor changes.

Minor points:

1: P9: The authors make the interesting observation that GFP-SRP19 and GFP-SRP72 were detected in 50% of all analyzed CBs. Can they speculate on whether this simply reflects detection sensitivity (i.e. the proteins are present in all CBs, but only cross the threshold for detection in some) or due to different classes of CBs?

2: P11: The authors note that nucleolar proteins amount "to close to a fifth (19%)" of the SRP14 interactome. Is this by mass or simply by number of proteins called as interactors?

3: The introduction is well written, but but reads like a quite extensive review.

4: Fig. 2C: "RNase" is partly cut-off.

5: As a general point, the term "associates with" is perhaps too vague as used; e.g. "SRP associates with scores of nucleolar proteins involved in ribosome biogenesis". Presumably only a very small fraction of these scores or proteins are directly associated with SRP. "Coprecipitates with..." might be more accurate.

Life Science Alliance manuscript #LSA-2024-02614-T

We thank the referees for their reports, comments, and useful suggestions that allowed us to improve greatly our manuscript. We found the comments very constructive, and we have responded specifically to them as follows.

After close reinspection of our data, we have replaced the IP-SILAC analysis of the SRP14 associations with a new one in the revised manuscript (Figure 5B) because the level of detection of the bait was not as high as in the other SILAC IP (\log_2 SILAC ratio around 6 for all the three other SRP proteins). With the new GFP-SRP14 SILAC-IP, the bait has a \log_2 SILAC ratio of 7, and a good number of partners are significantly enriched. It is important to note that the partners that are identified in this new SILAC IP are the same as in the previous analysis but there are also many additional partners that we have now included in our global analysis and new Figure 9 (network display).

The Venn diagram in Figure 6 has been modified accordingly.

The new mass spectrometry proteomics data for GFP-SRP14 has been deposited in the ProteomeXchange Consortium via the PRIDE partner repository and the dataset identifier (PXD051488) was revised in the Data Availability paragraph.

In the revised manuscript, one Figure (Figure 9) and one Table (Table S2) have been added (see below). Hence, the initial Table S2 is now Table S3.

Reviewer #1

Signal Recognition Particle (SRP) targets secreted and transmembrane proteins to the endoplasmic reticulum (ER). SRP assembly is only somewhat understood. The 7SL RNA is transcribed in the nucleoplasm and assembled with a series of protein factors in the nucleolus and cytoplasm as it is assembled into mature SRP in the cytoplasm. SRP partially assembles in the nucleolus and SRP deficiencies are associated with conditions that resemble ribosomopathies, however the relationship between the SRP, the nucleolus, and ribosome biogenesis has not yet been investigated. The authors explored this connection by generating several cell lines that inducibly express GFP-tagged SRP proteins, checking their ability to assemble into SRP particles, determining their subcellular localization, and performing SILAC-based proteomics that revealed association with nucleolar proteins. Further, the authors examined the dynamics of the SRP proteins upon nucleolar disruption. The authors show that two of the GFP-tagged proteins (SRP19 and SRP72) were able to efficiently assemble into mature SRPs, but two others (SRP9 and SRP14) were not. The latter two were used as representative proteins for complexes stalled in the early stages of SRP biogenesis. They found that SRP proteins associate with nucleolar proteins involved in ribosome biogenesis and discovered other novel proteins involved in SRP biogenesis (LYRIC and SND1). They demonstrated that the distribution of SRP proteins is altered upon nucleolar disruption with Actinomycin D, and re-discovered definitive colocalization of SRP proteins with Cajal bodies (see Discussion).

Overall, this paper explores the understudied and important connection between the nucleolus and SRP. The method to track the process of assembly with the subset of GFP-tagged SRP proteins stalled at early stages is a unique and interesting approach.

However, I am disappointed in the lack of scholarly use of the published literature on SRP in the nucleolus. For example, it is stated many times that it was "unexpected" that the SRP proteins associate with nucleolar proteins based on the SILAC experiments. They are also referred to as "newly uncovered." Yet the clear association of SRP proteins with nucleolar proteins is already available on numerous databases and has literally been posted there for years (see Biogrid and NCBI gene, for

examples.) This information should have been included in the Introduction for a comprehensive review of the literature. Furthermore, a comparison between the results reported here and prior results should be made in the Discussion. How well did prior results report nucleolar association of the SRP proteins? Are the results reported here really an advance, or do they tell us what we already knew but with higher precision because of SILAC? A comparison table should be made and put in the Supplementary Material, and then thoroughly analyzed in the Discussion. This group is not the first to find that nucleolar proteins associate with assembly SRP, and they should not give the impression that they are. It is not right.

Thank you. We completely agree with the reviewer that indeed some of the associations between SRP proteins and nucleolar proteins and/or proteins involved in ribosome biogenesis have been previously reported in protein/protein databases. These were unveiled in several high throughput analyses.

As suggested by the reviewer, we have added a sentence in the Introduction to address this (Page 4): “In agreement with this idea, a few interactions between SRP proteins and nucleolar proteins, including some known to be important for ribosome biogenesis have been reported (<https://thebiogrid.org>).”

We added a Table comparing what was known and what was not (Table S2).

Most importantly, we added a new Figure (Figure 9) to display in a color-coded network format the information that was available but required deep mining of published databases (about 1/4 of all listed interactions) and the completely new interactions (about 3/4).

We believe it is useful to bring to your attention, that to our knowledge, the associations available in the database were never discussed in any papers, and therefore remained completely ignored by the community.

As suggested by the reviewer, we have in the revised manuscript extensively compared our data with the data present in the literature in a new Figure (Figure 9) made with Cytoscape and in a new Table (Table S2A). We have only used data for *homo sapiens* reported in the Biogrid database (<https://thebiogrid.org>) since the data reported in NCBI are issued from Biogrid. We have selected data coming from IP experiments and analysis by mass spectrometry (Huttlin et al., 2015; 2017; 2021; Cho et al., 2022; Salvetti et al., 2016; Hein et al., 2015; Ewing et al 2007; Boldt et al., 2016; Jang et al., 2018; Liu et al., 2018), and excluded data issued from cell fractionations since in these cases the associations were not reliably confirmed.

We describe 2 different situations:

- We have discovered 239 new associations between SRP proteins and nucleolar proteins and/or proteins involved in ribosome biogenesis:

*161 new associations with 95 new nucleolar proteins and/or proteins involved in ribosome biogenesis (70 nucleolar proteins involved in ribosome biogenesis, 15 proteins with nucleolar localization but no function in ribosome assembly reporter yet, and 10 proteins involved in ribosome biogenesis in another compartment than the nucleolus) (Table S2A, newly discovered associations with new nucleolar proteins and/or proteins involved in ribosome biogenesis; Figure 9A, newly discovered associations, purple line).

* 79 new associations between SRP proteins and 53 nucleolar proteins and/or proteins involved in ribosome biogenesis that were already known to interact with other SRP proteins (Table S2A, newly discovered associations with nucleolar proteins and/or proteins involved in ribosome biogenesis that were already known to interact with other SRP proteins; Figure 9A, newly discovered associations, purple lines).

- We have confirmed 79 previously reported associations (Table S2A, confirmed associations; Figure 9A, confirmed associations, green lines).

Therefore, more than $\frac{3}{4}$ of the associations between nucleolar proteins and/or proteins involved in ribosome biogenesis with SRP proteins that we observed in our IP-SILAC experiments were never before reported. Our data brings to 173 the number of nucleolar proteins and/or proteins involved in ribosome biogenesis that are associated with SRP proteins (95 new proteins (circled in red in Figure 9A), and 78 confirmed proteins). This broad network of associations indicated a strong connection between SRP, ribosome biogenesis, and nucleolus.

Our experiments allowed the detection of new partners. This could be because we specifically induced the expression of the bait with doxycycline that would allow the enrichment of complexes with newly synthesized SRP proteins. Then, the factors bound during the assembly process of SRP particles would be better immunoprecipitated and identified.

In the revised manuscript, we have also compared our data with the data present in the literature concerning LYRIC, SND1, and the RQC components (Figure 9B and C; Table S2B).

Furthermore, the Introduction is too long, and contains extraneous paragraphs on topics that are not touched on further. It should be trimmed by at least a couple of paragraphs (4-5 total).

Thank you. The introduction has been trimmed substantially

I understand that it is difficult to probe what we all really want to know: what is the function of SRP assembly in the nucleolus? Given that that is too hard to do, overall the paper represents an important contribution through its exploration of the relationships between SRP, the nucleolus, and ribosome biogenesis if improvements are made as above and below.

Thank you.

Comments:

1. While this paper effectively demonstrate that an intact nucleolus is required for SRP biogenesis, the data do not show that the processes of ribosome biogenesis and SRP are coordinated as stated in the abstract. Can you please soften the language?

Thank you. With all due respect, we don't feel that we stated that "ribosome and SRP biogenesis are coordinated" in our Abstract. We simply suggested that it could be a possibility since we wrote that our data "suggest that the biogenesis of SRP and ribosomes may be coordinated". Therefore we left the sentence as it were.

2. For proteomics studies, the authors state that a percentage of the total nucleolar proteins (?) associated with the SRPs are nucleolar (see fig. 5). When these numbers were generated, was a cutoff value used to determine a positive association (i.e. the significance value)? The methodology for what constituted a hit and how these percentages were generated should be clarified.

In Figure 5, the indicated percentages of nucleolar proteins and/or proteins involved in ribosome biogenesis, as well as the one of ribosomal proteins, are the percentages in number of these classes of proteins among all the associated proteins with a given GFP-SRP protein. We have considered as a positive association, the associations with a SILAC ratio above 1 (see Table S1). In fact, 90% of the associations of SRP proteins with nucleolar proteins and/or proteins involved in ribosome biogenesis have a SILAC ratio above 2, and 80% have a SILAC ratio above 3.

We have added a sentence in the legend of Figure 5 to clarify these points (as well as point 2 of reviewer 2).

The new sentence reads as follows:

“The indicated percentage of nucleolar proteins and/or proteins involved in ribosome biogenesis, as well as the one of ribosomal proteins, represent the percentages in number of these classes of proteins among all the associated proteins with the GFP-SRP protein analyzed and with a SILAC ratio above 1.”.

We added also this sentence in the Material and Methods: “We have considered as a positive association, the associations with a SILAC ratio above 1.”.

3. Methodology for quantifying imaging data in figure 4 should be explained. How were the number of cells counted chosen? Were they counted manually, and if so by what criteria?

We observed manually all the CBs in all the cells that were present in the different images that we captured with the confocal microscope. We did not choose the number of cells or the number of CBs counted.

This sentence was added in the legend of Figure 4: “Counting was done manually by operators blinded to the samples observed”, to precise the method.

Reviewer #2

The authors report analyses that address the long-standing observation of SRP components within the nucleolus. Tagged constructs are used to follow the localization and assembly status of ectopically expressed SRP proteins. This confirmed nucleolar localization and interactions, and provided evidence for links between nucleolar structure and SRP assembly. In addition, SRP proteins were identified in a subset of CBs, indicating their possible involvement in specific assembly steps.

Overall, the novel insights are limited, but the analyses appear to have been well-performed, while the MS is well written and will be of value and interest to the field. I would support publication with only minor changes.

Thank you for an overall positive assessment.

With all due respect, we would like to disagree with the notion that ‘our work contributes limited novel insights’ in the field.

In this work:

-We have identified 95 new nucleolar partners of SRP, with $\frac{3}{4}$ of interactions being entirely new, and $\frac{1}{4}$ requiring deep mining of databases populated with (often noisy) high throughput screens datasets (listing interactions never discussed in the context of SRP biology).

-We have identified formally Cajal bodies as a novel putative site of SRP assembly.

-We have demonstrated the importance of a functional nucleolus for SRP biogenesis.

-We have localized SRP components within the PDFC, a sub-nucleolar compartment that is only known for a few months.

Minor points:

1: P9: The authors make the interesting observation that GFP-SRP19 and GFP-SRP72 were detected in 50% of all analyzed CBs. Can they speculate on whether this simple reflects detection sensitivity (i.e.

the proteins are present in all CBs, but only cross the threshold for detection in some) or due or different classes of CBs?

Thank you for this very interesting remark, which is extremely challenging to address experimentally. At this stage, we cannot exclude one of these two possibilities. Therefore, we added in the Result section (Page 9): “Either they are two classes of CBs, containing or not SRP proteins, or SRP proteins are present in all CBs but the experimental conditions used did not allow to detect SRP proteins in some of them.”

2: P11: The authors note that nucleolar proteins amount "to close to a fifth (19%)" of the SRP14 interactome. Is this by mass or simply by number of proteins called as interactors?

This is a percentage 'by number' of proteins. We added a sentence in the legend of Figure 5 to clarify this point (as well as point 2 of reviewer 1): “The indicated percentage of nucleolar proteins and/or proteins involved in ribosome biogenesis, as well as the one of ribosomal proteins, represent the percentages in number of these classes of proteins among all the associated proteins with the GFP-SRP protein analyzed and with a SILAC ratio above 1.”.

3: The introduction is well written, but reads like a quite extensive review.

The introduction has been streamlined and is now more compact. Several paragraphs have been removed.

4: Fig. 2C: "RNase" is partly cut-off.

We have corrected this point. We have also replaced “RNase” with “RNase”.

5: As a general point, the term "associates with" is perhaps too vague as used; e.g. "SRP associates with scores of nucleolar proteins involved in ribosome biogenesis". Presumably only a very small fraction of these scores or proteins are directly associated with SRP. "Coprecipitates with..." might be more accurate.

Thank you. We appreciate this comment. We believe it all comes down to semantics, which is all very important in Biology and, in this case, to personal preferences. We believe the nature of our dataset, and how it was generated (SILAC) and confirmed (for some interactions, by pull down) is amply described in our manuscript to not mislead any readers. Therefore, we would like to suggest keeping the term « associates », which we believe is well put in context at all places in our text.

May 23, 2024

RE: Life Science Alliance Manuscript #LSA-2024-02614-TR

Dr. Severine Massenet
University of Lorraine
CNRS - UMR7365
avenue de la forêt de Haye, BP 184
Vandoeuvre-les-Nancy, Cedex 54505
France

Dear Dr. Massenet,

Thank you for submitting your revised manuscript entitled "The Nucleolar Phase of Signal Recognition Particle Assembly". We would be happy to publish your paper in Life Science Alliance pending final revisions necessary to meet our formatting guidelines.

- please be sure that the authorship listing and order is correct
- please upload all figure files as individual ones, including the supplementary figure files; all figure legends should only appear in the main manuscript file
- please add the Twitter handle of your host institute/organization as well as your own or/and one of the authors in our system
- please remove tracked changes from the manuscript file
- please incorporate any points from the Conclusions section into the Discussion
- please add your main, supplementary figure, and table legends to the main manuscript text after the references section
- please add callouts for Figure 5A-B; S2A-D and S3A-B to your main manuscript text

FIGURE CHECKS:

- please add sizes next to all blots

LSA now encourages authors to provide a 30-60 second video where the study is briefly explained. We will use these videos on social media to promote the published paper and the presenting author (for examples, see <https://docs.google.com/document/d/1-UWCfbE4pGcDdcgzcmiuJl2XMBJnxKYeqRvLLrLS08s/edit?usp=sharing>). Corresponding or first-authors are welcome to submit the video. Please submit only one video per manuscript. The video can be emailed to contact@life-science-alliance.org

A. FINAL FILES:

B. MANUSCRIPT ORGANIZATION AND FORMATTING:

Sincerely,

May 28, 2024

RE: Life Science Alliance Manuscript #LSA-2024-02614-TRR

Dr. Severine Massenet
University of Lorraine
CNRS - UMR7365
Avenue de la foret de Haye, BP 184
Vandoeuvre-les-Nancy, Cedex 54505
France

Dear Dr. Massenet,

Thank you for submitting your Research Article entitled "The Nucleolar Phase of Signal Recognition Particle Assembly". It is a pleasure to let you know that your manuscript is now accepted for publication in Life Science Alliance. Congratulations on this interesting work.

DISTRIBUTION OF MATERIALS:

Again, congratulations on a very nice paper. I hope you found the review process to be constructive and are pleased with how the manuscript was handled editorially. We look forward to future exciting submissions from your lab.

Sincerely,
